# BRE/BRCC45 regulates CDC25A stability by recruiting USP7 in response to DNA damage

Kajal Biswas[1], Subha Philip[1,3], Aditya Yadav[1], Betty K. Martin[1,2], Sandra Burkett[1], Vaibhav Singh[1], Anav Babbar[1], Susan Lynn North[1], Suhwan Chang[1,4] & Shyam K. Sharan[1]

BRCA2 is essential for maintaining genomic integrity. BRCA2-deficient primary cells are either not viable or exhibit severe proliferation defects. Yet, *BRCA2* deficiency contributes to tumorigenesis. It is believed that mutations in genes such as *TRP53* allow *BRCA2* heterozygous cells to overcome growth arrest when they undergo loss of heterozygosity. Here, we report the use of an insertional mutagenesis screen to identify a role for BRE (*Brain and Reproductive organ Expressed*, also known as BRCC45), known to be a part of the BRCA1-DNA damage sensing complex, in the survival of BRCA2-deficient mouse ES cells. Cell viability by BRE overexpression is mediated by deregulation of CDC25A phosphatase, a key cell cycle regulator and an oncogene. We show that BRE facilitates deubiquitylation of CDC25A by recruiting ubiquitin-specific-processing protease 7 (USP7) in the presence of DNA damage. Additionally, we uncovered the role of CDC25A in BRCA-mediated tumorigenesis, which can have implications in cancer treatment.

[1] Mouse Cancer Genetics Program, Center for Cancer Research, National Cancer Institute, Frederick, MD 21702, USA. [2] Leidos Biomedical Research, Inc., Frederick National Laboratory for Cancer Research, Frederick, MD 21702, USA. [3]Present address: Maxim Biomedical Inc., Rockville, MD 20850, USA. [4]Present address: Department of Biomedical Sciences, Department of Physiology, University of Ulsan School of Medicine, Seoul 05505, South Korea. Correspondence and requests for materials should be addressed to S.K.S. (email: sharans@mail.nih.gov)

nherited mutations in *BRCA2* are not only associated with increased risk of breast and ovarian cancer but also prostate, pancreatic, and other cancers[1–6]. BRCA2 is required for RAD51-mediated repair of DNA double strand breaks (DSB) by homologous recombination (HR) as well as for protecting stalled replication forks[7–9]. BRCA2-deficient cells undergo tumorigenesis due to their unstable genome[10]. Paradoxically, loss of *BRCA2* in normal cells leads to cell cycle arrest and apoptosis due to the activation of the DNA damage response (DDR) rather than unrestrained proliferation, characteristic of cancer cells[11,12]. It has been postulated that mutation in genes such as *TRP53* contributes to the survival of *BRCA2*-deficient cells. In mice, the lethality of *Brca2*-null embryos can be partially rescued on a *Trp53*-deficient genetic background and significant enhancement of tumorigenesis was observed when *Brca2* loss was combined with a *Trp53* mutation in mice[13,14]. Although mutations in *TRP53* have been identified in tumors from *BRCA2* mutation carriers[15,16], it is not inactivated in all *BRCA2*-deficient tumors, suggesting the existence of alternative routes to overcome growth arrest of *BRCA2*-deficient cells. The evidence for the existence of such *BRCA2*-cooperating genes to promote tumorigenesis is very limited. We have recently shown that *Parp1* heterozygosity can contribute to the viability of *Brca2*-null cells as well as tumorigenesis in *Brca2* mutant mice[17]. In the current study, we have undertaken an insertional mutagenesis approach using Murine Stem Cell Virus (MSCV) to identify novel genes that can support the survival of *BRCA2*-deficient cells. Recently, we identified upregulation of *Gipc3* to rescue the viability of *Brca2*-null ES cells using a similar approach[18].

Our MSCV-based genetic screen resulted in the identification of BRE that can support the viability and growth of *Brca2*-null cells by regulating cell cycle checkpoint protein CDC25A. *BRE*, an evolutionarily conserved stress-modulating gene, is upregulated in hepatocellular carcinoma and has been identified as a risk-modifier of BRCA1-associated ovarian cancer[19,20]. BRE was found to be a part of the BRCA1-A and multiprotein BRISC complex (BRCC36 isopeptidase complex) that specifically cleaves K-63 polyubiquitin chains in various substrates[21,22]. Interaction of BRE with MERIT40 is required to maintain the integrity of BRCA1-A complex[23]. BRE and BRCC36 interaction is also important for E3 ligase activity of BRCA1–BARD1 heterodimer and K-63 deubiquitylase activity of BRCC36[22,24]. BRE knockdown disrupts BRCA1 foci formation at DNA damage sites and thus plays an important role in DNA repair[23,25]. It is known to interact with Fas and TNF-R1 and acts as a death receptor-associated anti-apoptotic protein[26,27]. *Bre*-null mice are viable and fertile and exhibit no overt phenotype[28]. However, fibroblasts derived from mutant mice show proliferation defect, premature senescence, and a defect in DSB repair by HR[28].

Here, we show that BRE regulates the levels of CDC25A in response to DNA damage by promoting its deubiquitylation via its interaction with ubiquitin-specific-processing protease 7 (USP7). The BRE–CDC25A–USP7 complex is a multiprotein complex that is distinct from the well-established BRE containing BRISC or BRCA1-A complex. We further show that this new function of BRE in CDC25A regulation plays an important role in the survival of BRCA1/2-deficient cells. Importantly, we found higher expression of CDC25A in BRCA1/2-deficient tumors in different breast cancer data sets. Furthermore, we confirmed a positive correlation between BRE and CDC25A levels in human breast tumors. Overall, we demonstrate that CDC25A is a critical factor in the survival and growth of BRCA1/2-deficient cells and BRE, an adapter protein, plays an important role in regulating the CDC25A protein levels by recruiting USP7.

## Results

**Genetic screen to identify *BRCA2* genetic interactors.** To identify the genes that may cooperate with *Brca2* in the process of tumorigenesis by contributing to cell viability, we performed an MSCV-based insertional mutagenesis screen in mES cells. We hypothesized that any mutation due to the viral insertion that supports the viability of *Brca2KO/KO* cells can be a potential genetic interactor. We used MSCV because its strong long terminal repeats (LTRs) are known to be active in several mammalian cell lines including ES cells and can induce the expression of neighboring genes[18,29]. Viral insertion can also disrupt genes. We used the previously reported PL2F7 mES cells that have one conditional allele of *Brca2* (CKO) and the other allele is functionally null (KO) to perform the mutagenesis screen[6]. In these cells, CRE-mediated deletion of the conditional allele generates a functional *HPRT* minigene that allows selection of recombinant clones. Because BRCA2 is essential for cell viability, no viable *Brca2KO/KO* ES cells are obtained in hypoxanthine-aminopterin-thymidine (HAT) media after CRE-mediated recombination in PL2F7 cells (Fig. 1a)[6]. However, when PL2F7 cells were transduced with MSCV-CRE (MSCV expressing Cre), *Brca2KO/KO* HAT-resistant colonies were obtained that had one or two viral integrations (Fig. 1b, Supplementary Fig. 1A).

**BRE overexpression rescues *Brca2KO/KO* mES cell's lethality.** To identify the viral insertion sites, we used a splinkerette polymerase chain reaction (PCR)-based method[30]. One of the viral integrations (in Clone 3d) was at chromosome 5qB1. The integration was within the *Mitochondrial Ribosomal Protein L33* (*Mrpl33*) and upstream of *Ribokinase* (*Rbks*) and *Brain and Reproductive organ Expressed* (*Bre*) genes (Supplementary Fig. 1B). In addition, within the *Rbks* locus microRNA 3473e (*Mir3473e*) is also present that has no human ortholog. To determine the genes that are affected by viral insertion, we performed real-time reverse transcriptase (RT) PCR and compared the expression levels of *Mrpl33*, *Rbks*, and *Bre* in PL2F7 cells and *Brca2KO/KO;Clone 3d* cells. Among these genes, only *Bre* mRNA showed a significant upregulation (~1.8-fold higher) in *Clone 3d* cells (Fig. 1c and Supplementary Fig. 1C).

To test whether the overexpression of *Bre* can account for the viability of *Clone 3d*, we expressed *Hemagglutinin* (*HA*)-tagged *BRE* cDNA under the control of *MSCV LTR* in PL2F7 cells (Fig. 1d). CRE was expressed in two independent HA-BRE expressing clones to delete the conditional allele (Fig. 1e, top and middle panels). The HAT-resistant clones were then genotyped to identify the clones that have lost conditional *Brca2* allele. We obtained *Brca2KO/KO* mES cells (referred as *Brca2KO/KO; MSCV-BRE*) at a frequency of 8–10% (percentage of HAT-resistant recombinant clones) in both the clones tested (Fig. 1e). The clones that do not overexpress BRE failed to generate any viable *Brca2KO/KO* mES cells after CRE expression (Fig. 1e, lower panel).

We next tested whether BRE overexpression can promote the growth of human BRCA2-deficient cells. We transduced MCF7 cells with (MCF7BRE) or without (MCF7Neo) BRE overexpression with lentiviruses expressing two different shRNAs targeting BRCA2 and a non-specific shRNA (Fig. 1f). As predicted, MCF7Neo cells showed a significantly slower growth in response to BRCA2 knockdown (Fig. 1f). In contrast, BRE overexpression rescued this growth defect in MCF7BRE cells, which further supports its role in promoting the growth of BRCA2-deficient cells (Fig. 1f).

***Bre* overexpressing *Brca2KO/KO* cells have DNA repair defects.** We examined whether *Bre* overexpression could compensate for the loss of *Brca2* and restore those functions in *Brca2KO/KO;*

MSCV-BRE ES cells[31,32]. We tested $Brca2^{KO/KO}$; MSCV-BRE cells for their ability to form RAD51 foci in response to ionizing radiation (IR) and also examined the integrity of replication forks in response to hydroxyurea (HU). We found that $Brca2^{KO/KO}$; MSCV-BRE ES cells are defective in both IR-induced RAD51 foci formation (Fig. 2a, b, Supplementary Fig. 2A) as well as protection of stalled replication forks (Fig. 2c). Further, we tested those cells for sensitivity to various genotoxins and found them to be hypersensitive to all DNA-damaging agents tested (Fig. 2d, e and Supplementary Fig. 2B-D). When we examined the metaphase spreads of these cells, we found an increase in chromosomal aberrations (Fig. 2f, g). Taken together, these results suggest that BRE overexpression does not restore the BRCA2-loss-associated defects.

**Bre overexpressing cells show radio-resistant DNA synthesis.** The embryonic lethality of Brca2-null mice can be partially rescued by targeted deletion of Trp53[13]. We tested whether the Brca2-deficient ES cells rescued by BRE overexpression have intact TRP53 regulation in response to DNA damage. Increased expression of TRP53 in these cells in response to IR suggests that

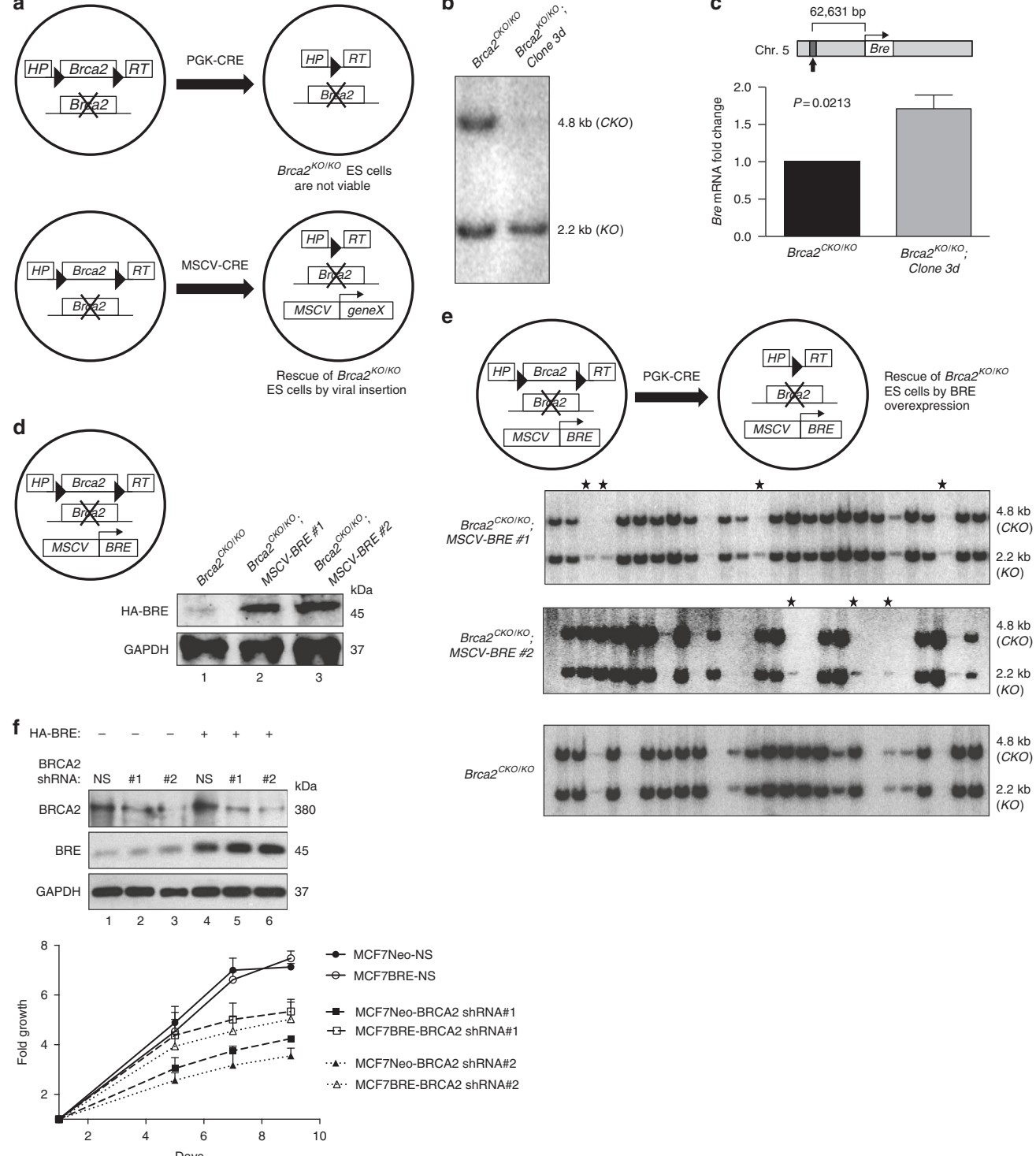

BRE-mediated survival of $Brca2^{KO/KO}$ cells is independent of TRP53 (Supplementary Fig. 2E). We then investigated whether the survival of $Brca2$-deficient mES cells by $Bre$ overexpression is due to a defect in the control of cell cycle progression after DNA damage. We first checked the progression of cells from G1 to S phase after IR by staining the cells with 5-bromo-2′-deoxyuridine (BrdU) and propidium iodide (PI). We found that both the control and $Brca2$-deficient cells entered S phase after IR, indicating a relaxed G1/S checkpoint in mES cells (Supplementary Fig. 2F-G), consistent with previous reports[33–36]. We then measured the G2/M checkpoint by staining for the DNA content and phosphorylated histone H3, a mitotic marker[37]. Cells with a functional BRCA2 arrest in the G2 phase immediately after IR even when $BRE$ is over expressed (Supplementary Fig. 2H-I). The cells lacking BRCA2 ($Brca2^{KO/KO}$; $MSCV-BRE$ or $Brca2^{KO/KO}$; $Clone\ 3d$) exhibited a minor defect in G2/M checkpoint (Supplementary Fig. 2H-I). This defect in the G2/M checkpoint is a feature of $Brca2$-deficient cells[38].

We analyzed $Bre$ overexpressing cells for intra S-phase checkpoint activation by the radio-resistant DNA synthesis assay. When S-phase cells are exposed to IR, the intra S-phase checkpoint is activated and inhibits DNA synthesis. This inhibition can be measured by [3]H-thymidine incorporation[39]. The results indicate that the $Brca2^{CKO/KO}$ cells slowed DNA synthesis following DNA damage (Fig. 3a). In contrast, all $BRE$ overexpressing cells ($Brca2^{CKO/KO}$; $MSCV-BRE$, $Brca2^{KO/KO}$; $Clone\ 3d$ or $Brca2^{KO/KO}$; $MSCV-BRE$) continued DNA synthesis (Fig. 3a). This suggests that the defect in inhibition of DNA synthesis after IR is not dependent on the absence of functional $Brca2$, rather dependent on $BRE$ overexpression. After DNA damage, inhibition of DNA synthesis is essential to repair the damage. $BRE$ overexpression impairs this inhibition of DNA synthesis after DNA damage and this impairment might help the BRCA2-deficient cells to continue the cell cycle.

**Defects in IR-induced CDC25A degradation by BRE overexpression.** The critical event in the inhibition of DNA synthesis following DNA damage is accelerated proteolysis of cell division cycle A (CDC25A) protein, a dual specificity protein phosphatase and one of the most crucial cell cycle regulators[40]. DNA damage-induced degradation of CDC25A protein in turn leads to the inhibition of the cyclin dependent kinase (CDK) activity that facilitates transient blockade of DNA replication[40]. To check the CDC25A level in BRE overexpressing cells, we monitored the protein level following IR in $Brca2^{CKO/KO}$, $Brca2^{CKO/KO}$; $MSCV-BRE$, $Brca2^{KO/KO}$; $Clone\ 3d$ and $Brca2^{KO/KO}$; $MSCV-BRE$ cells at different time points. The results show that the level of CDC25A protein is comparable among the untreated BRE normal and overexpressing cells (Fig. 3b and Supplementary Fig. 3, lanes 1, 5). Furthermore, a reduction in CDC25A is rapidly induced in control cell line ($Brca2^{CKO/KO}$) in response to IR (Fig. 3b and Supplementary Fig. 3). However, BRE overexpressing cells maintain higher CDC25A levels at all time points after IR compared to the corresponding time points in control cells (Fig. 3b and Supplementary Fig. 3A-B). This indicates that the IR-induced CDC25A reduction is compromised upon BRE overexpression. Next, we tested how this elevated level of CDC25A in response to IR affects the level of phosphorylated CDK1 (inactive). While the phosphorylated CDK1 levels increased after IR in the control cells, BRE overexpressed cells maintained their CDK1 levels similar to the untreated cells (Fig. 3b). These data confirm that blocking the CDC25A decrease in response to IR can activate its target protein to help these cells to proceed into cell cycle even after DNA damage.

**Regulation of CDC25A by BRE in human cancer cell lines.** We next tested whether BRE-mediated stabilization of CDC25A levels in response to IR occurs in MCF7 cells. To stably overexpress, we used a $loxP-STOP-loxP$ cassette to induce the expression of the $BRE$ transgene under the control of the chicken actin (CAG) promoter after CRE expression. The levels of CDC25A were not altered upon BRE overexpression in unirradiated MCF7 cells (Fig. 3c, lanes 1, 4; $P = 0.21$). However, after IR, while CDC25A levels were reduced in parental MCF7 cells (Fig. 3c, lanes 2, 3), cells overexpressing BRE did not show any change in CDC25A level (Fig. 3c, lanes 5, 6). These results suggested that the effect of BRE overexpression on CDC25A stabilization occurred in other cells as well.

To examine the effect of endogenous BRE on CDC25A regulation, we knocked down BRE in MCF7 cells using two different siRNAs and assessed IR-induced CDC25A degradation. As shown in Fig. 3d, a reduction in BRE levels clearly resulted in accelerated degradation of CDC25A (Fig. 3d, compare lane 2–8 with lanes 10–16 and 18–24). We further evaluated the effect of BRE knockdown on CDC25A degradation in other human cancer cell lines. We screened NCI-60 cell panel for BRE and CDC25A expression and selected prostate cancer cell line PC3 and colorectal carcinoma cell line SW620 for higher BRE and CDC25A expression. We also tested a breast cancer cell line ZR75-1 due to its higher level of CDC25A[41]. We found that BRE knockdown caused increased IR-induced CDC25A degradation in all three cell lines (Supplementary Fig. 4, compare lane 3 with lanes 6, 9, and 12 in each panel).

BRE protein stabilization is dependent on MERIT40, a physical interactor of BRE and a component of BRCA1 complex[23]. Therefore, we examined whether the knockdown of MERIT40 had any effect on CDC25A stability in BRE overexpressing cells.

**Fig. 1** Identification of BRE as a genetic interactor of BRCA2 using a MSCV-based insertional mutagenesis screen. **a** Schematic representation of MSCV-mediated insertional mutagenesis in $Brca2$ conditional mouse ES cells. $Brca2^{KO/KO}$ cells generated after PGK-CRE-mediated deletion of the conditional allele are not viable. Mutagenesis by MSCV-CRE can generate viable $Brca2^{KO/KO}$ ES cells. **b** Southern blot analysis of HAT-resistant ES cell colony that lost conditional $Brca2$ allele ($Brca2^{KO/KO}$; $Clone\ 3d$) after MSCV-CRE transduction. Upper band: conditional allele (CKO); lower band: knock-out allele (KO). **c** Quantification of $Bre$ expression by real-time RT-PCR in $Brca2$ conditional mutant ($Brca2^{CKO/KO}$) and $Brca2^{KO/KO}$ ES cells with viral insertion at chr: 5qB1 ($Brca2^{KO/KO}$;$Clone\ 3d$). Data are represented as mean ± s.d. Top panel shows the distance between viral insertion and start of first exon of $Bre$. **d** Expression of HA-tagged BRE from $MSCV\ LTR$ in $Brca2^{CKO/KO}$ clones. Two independent clones Clone #1 and Clone #2 analyzed by western blot analysis were used further. Left panel shows the scheme of relevant alleles of $Brca2^{CKO/KO}$; $MSCV-BRE$ ES cells. **e** Southern blot analysis of HAT-resistant ES cell colonies after CRE-mediated deletion of conditional allele in $Brca2^{CKO/KO}$; $MSCV-BRE$ ES cells to identify $Brca2^{KO/KO}$ clones (marked with solid stars), upper band: conditional allele (CKO); lower band: knock-out allele (KO). Schematic diagram of CRE-induced loss of conditional $Brca2$ allele in $Brca2^{CKO/KO}$; $MSCV-BRE$ cells is shown at the top. **f** Upper panel shows western blot for BRCA2 knockdown by two different shRNAs (#1 and #2) and a non-specific (NS) control and HA-BRE expression in MCF7 cells that were stably expressing either empty vector (MCF7Neo) or vector expressing HA-BRE (MCF7BRE). GAPDH was used as a loading control. Growth of MCF7 cells after BRCA2 knockdown in the presence or absence of HA-BRE expression represented in the lower panel. Fold growth was calculated by dividing cell counts on particular day with cell count on day 1. $P$ values are shown in Supplementary Table 2. $P$ values were calculated using paired two-tailed $t$-test

After depletion of MERIT40 with siRNA, we failed to get the overexpression of BRE and in those cells, CDC25A undergoes degradation in response to IR (Fig. 3e, lanes 7–12). These data further support a role for BRE in CDC25A stabilization after DNA damage.

**CDC25A is post-translationally stabilized by BRE**. We examined whether BRE transcriptionally regulates *CDC25A* levels. However, we did not observe any significant changes in *CDC25A* mRNA levels in cells overexpressing BRE (Supplementary Fig. 5A). To examine if BRE overexpression affected IR-induced

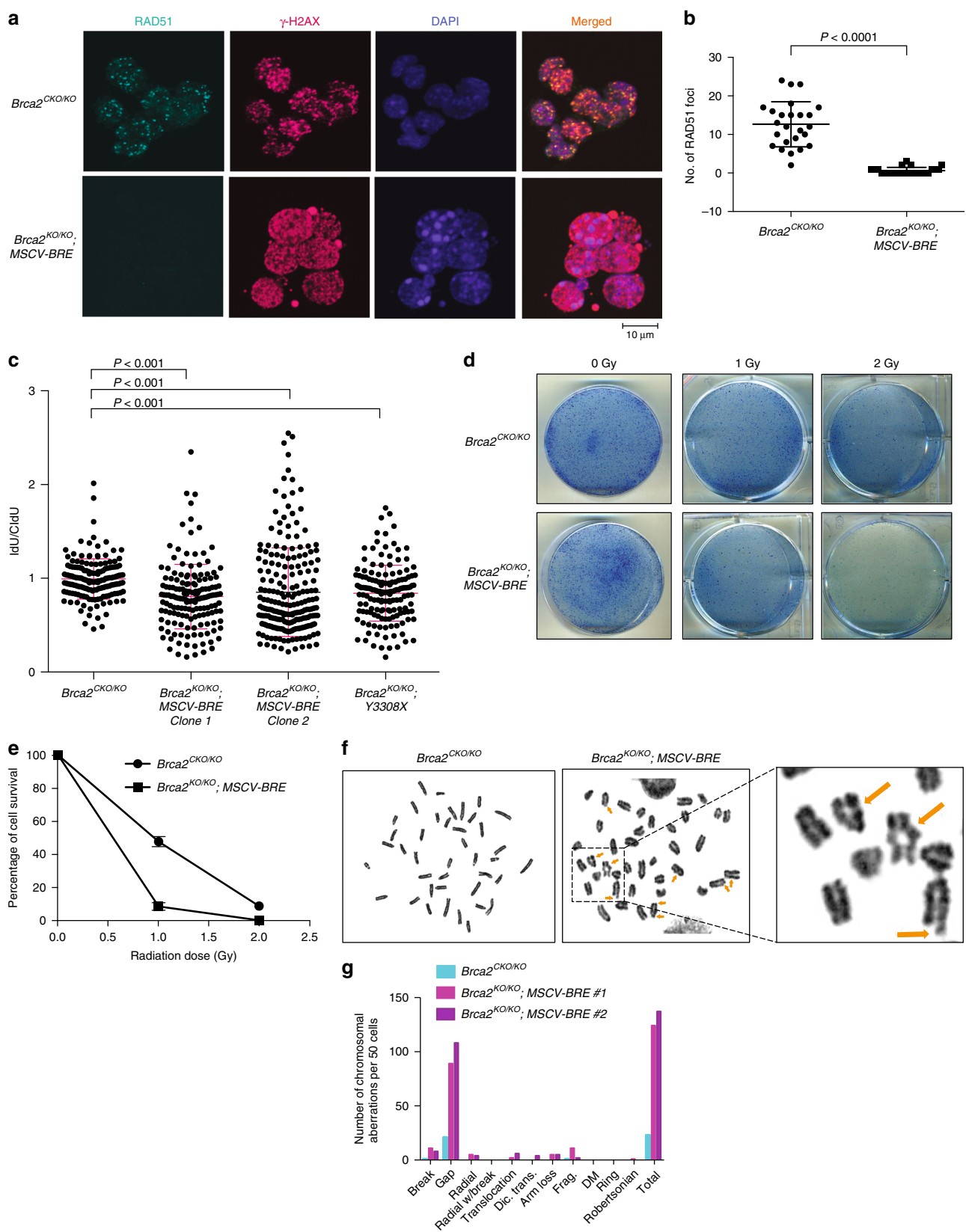

CDC25A degradation, we analyzed the level of CDC25A protein in the presence of cycloheximide, an inhibitor of protein translation. Interestingly, we found that BRE overexpression enhanced the half-life of CDC25A in response to IR (Supplementary Fig. 5B-C).

Upon exposure to genotoxic stress, CDC25A undergoes ubiquitin-mediated proteolysis[42,43]. Therefore, we examined whether BRE overexpression affected ubiquitylation of CDC25A after IR. To facilitate the detection of CDC25A, we transiently overexpressed Flag-tagged CDC25A. For equal detection of CDC25A in untreated and irradiated samples, cells were irradiated in the presence of the proteasome inhibitor MG132 and were then processed for analysis of CDC25A polyubiquitylation. Although BRE overexpression did not change the ubiquitylation level in unirradiated cells (Fig. 4a, lanes 1, 5), it resulted in significant reduction in the amount of ubiquitylated CDC25A in irradiated cells compared to control cells (Fig. 4a, lanes 2, 6). We further confirmed that the BRE overexpression affects CDC25A K-48 linked poly-ubiquitylation and not K-63 linked (Supplementary Fig. 6).

**BRE mediates CDC25A stabilization after DNA damage by USP7.** DNA damage induces phosphorylation of CDC25A in a CHK1-dependent manner[44]. Phosphorylated CDC25A is recognized and bound by SCF (Skp, Cullin, F-box) E3-ligase β-TRCP (β-transducin repeat-containing protein) that mediates subsequent ubiquitylation and proteolysis of CDC25A after DNA damage[45]. To determine if BRE overexpression affected CDC25A phosphorylation or CHK1 activation, we compared phosphorylated CDC25A levels and phosphorylated Ser317-CHK1 before and after irradiation. We observed that BRE overexpression neither affected phosphorylation of CDC25A (Supplementary Fig. 7A) nor phosphorylated Ser317-CHK1 (Supplementary Fig. 7B). Furthermore, we found that the level of β-TRCP is not altered upon BRE overexpression and exposure to IR (Supplementary Fig. 7C). We also assessed the level of ubiquitin hydrolase DUB3 that is known to deubiquitylate and stabilize CDC25A[46]. DUB3 levels remained unaltered in MCF7 cells overexpressing BRE compared to parental MCF7 cells, before and after IR (Supplementary Fig. 7C). We also failed to detect any physical interaction between BRE and DUB3 (Supplementary Fig. 7D) or β-TRCP (Supplementary Fig. 7E), indicating that BRE overexpression-mediated CDC25A deregulation is independent of β-TRCP and DUB3.

Next, we searched for other DUBs (deubiquitylating enzymes) that are known to interact with BRE to determine the mechanism by which BRE overexpression reduces CDC25A degradation in response to DNA damage. BRE is a part of the BRISC complex (BRCC36 isopeptidase complex) where it interacts with BRCC36, a DUB that selectively cleaves K63-linked polyubiquitin[21,22,47]. K48 polyubiquitin is the major signal for protein degradation, while K63 ubiquitin fulfills other roles such as endocytic trafficking, protein localization after DNA damage response, and activation of the nuclear factor-κB and T-cell receptor pathways in mammalian cells[48]. This suggests that BRCC36 may

not play a role in BRE-mediated CDC25A stability. Furthermore, we failed to detect any interaction between BRCC36 and CDC25A making it unlikely to be involved in CDC25A regulation (Supplementary Fig. 7F).

A global proteomic analysis of DUBs and their associated protein complexes identified an interaction between BRE and USP7[49]. USP7 controls the ubiquitylation of many key proteins such as HDM2, FOXO4, PTEN, and Claspin and plays a major role in regulating genome stability and cancer prevention (reviewed in ref. [50]). To investigate if BRE and USP7 interaction plays a role in the stabilization of CDC25A after IR, we knocked down USP7 in BRE overexpressing and control MCF7 cells using two different siRNAs (Fig. 4b). USP7 knockdown did not change CDC25A levels in unirradiated cells either in presence or absence of overexpressed BRE (Fig. 4b, lanes 1, 3, 5, and 7, 9, 11). In BRE overexpressing cells, USP7 knockdown resulted in IR-induced CDC25A degradation indicating that USP7 plays a key role in the stabilization of CDC25A after DNA damage in those cells (Fig. 4b, lanes 7–12). In contrast to CDC25A, the levels of HDM2, a known USP7 substrate, were reduced independent of IR or BRE overexpression[51] (Fig. 4b).

We also evaluated the effect of USP7 inhibitor P5091 on IR-induced CDC25A degradation (Supplementary Fig. 8)[52]. In the absence of BRE overexpression, CDC25A was degraded in response to IR independent of the inhibitor treatment (Supplementary Fig. 8, lanes 1–4). However, in BRE overexpressing cells, while CDC25A did not undergo degradation after irradiation, its levels were reduced in the presence of USP7 inhibitor (Supplementary Fig. 8, lanes 5–8). This further confirms the involvement of USP7 in BRE-mediated suppression of IR-induced CDC25A degradation.

Next, we examined CDC25A polyubiquitylation in the presence and absence of BRE upregulation after USP7 knockdown. To detect polyubiquitylation, we treated the cells with MG132 after IR. We observed that USP7 knockdown in BRE overexpressing cells resulted in an increase in the level of polyubiquitylated CDC25A after IR (Supplementary Fig. 9A, lanes 9, 10, 13, 14) when compared to controls (Supplementary Fig. 9A, lanes 5 and 6). These data support that BRE upregulation facilitates deubiquitylation of CDC25A in the presence of DNA damage by USP7. We next evaluated the effect of overexpression of USP7 and a catalytic inactive USP7 on CDC25A polyubiquitylation (Fig. 4c)[53]. Similar to BRE, overexpression of USP7 reduces IR-induced polyubiquitylation of CDC25A (Fig. 4c, lanes 4 and 5) whereas, polyubiquitylated CDC25A is observed after IR when catalytic inactive USP7 is expressed (Fig. 4c, lanes 6 and 7). However, restoration of polyubiquitylated CDC25A levels is incomplete, possibly due to the presence of endogenous USP7. Taken together, these results further support a role for USP7 in CDC25A regulation.

**BRE, USP7, and CDC25A complex formation upon DNA damage.** To examine whether USP7 regulates CDC25A via direct physical interaction or by some other mediator protein, we performed immunoprecipitation using different antibodies (Fig. 4d

**Fig. 2** BRE overexpression rescues Brca2[KO/KO] cell viability without affecting BRCA2 function. **a** Ionizing radiation (IR)-induced RAD51 foci formation in Brca2[CKO/KO] and Brca2[KO/KO]; MSCV-BRE ES cells. RAD51 foci are in turquoise, γ-H2AX foci in magenta, and nuclei are stained with 4,6-diamidino-2-phenylindole (DAPI, blue). Merged images show co-localization of RAD51 and γ-H2AX foci (yellow foci). **b** Quantification of RAD51 foci shown in **a**. Twenty-five nuclei were counted for each genotype. Error bars represent the mean ± s.d. **c** Scatter plot showing DNA fiber analysis of the indicated cell lines. P values were calculated using Mann–Whitney test. **d** Methylene blue-stained plates showing the sensitivity of Brca2[CKO/KO] and Brca2[KO/KO]; MSCV-BRE ES cells to different doses of IR. **e** Quantification of the colonies shown in **d**. Colony numbers obtained in plates with no treatment (0 Gy) are considered as 100%. Average of three experiments is shown. Error bars represent s.d. values. **f** Representative metaphase spread showing chromosomal aberration (marked with arrows) in Brca2[CKO/KO] and Brca2[KO/KO]; MSCV-BRE ES cells. **g** Quantification of different aberrations as well as total aberrations in 50 randomly selected Brca2[CKO/KO] and Brca2[KO/KO]; MSCV-BRE metaphase spreads

and Supplementary Fig. 9B–C). Immunoprecipitation with BRE antibody pulled down CDC25A independent of IR, but in this complex, endogenous USP7 was detected only after IR (Fig. 4d). BRE overexpression resulted in an increase in the amount of immunoprecipitated CDC25A and USP7, indicating that BRE overexpression facilitates increased interaction between BRE-CDC25A-USP7 in response to IR (Fig. 4d). Immunoprecipitation using USP7 antibodies further confirmed that this interaction is induced only after DNA damage (Supplementary Fig. 9B). Finally, we confirmed the interaction between BRE, CDC25A, and USP7 by pulling down Flag-tagged CDC25A in irradiated

cells that also overexpresses BRE (Supplementary Fig. 9C). In Flag-tagged CDC25A immunoprecipitation complexes, we failed to detect endogenous USP7 in irradiated control cells (Supplementary Fig. 9C). This is likely due to our inability to detect very low levels of endogenous USP7 by western blotting. Further, we tested whether the interaction between USP7 and CDC25A is dependent on BRE. We knocked down BRE in MCF7 cells and failed to detect USP7/CDC25A interaction, indicating that USP7 interaction with CDC25A depends on BRE (Supplementary Fig. 9D). We also evaluated if some of the known BRE interactors are present in BRE/CDC25A complex. We failed to co-

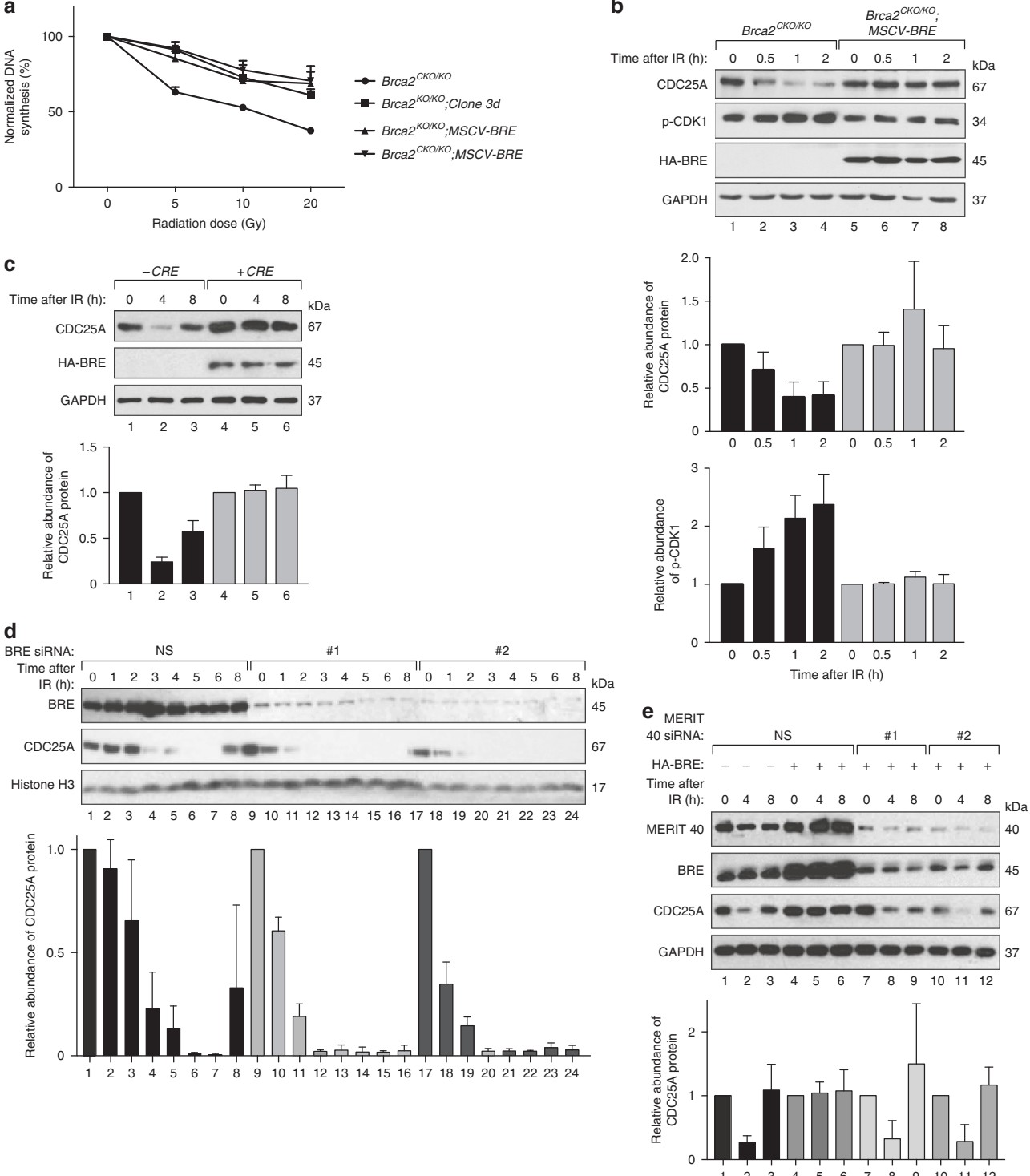

immunoprecipitate BRCC36, BRCA1, or RAP80 with CDC25A, whereas MERIT40 was found to interact with CDC25A (Supplementary Fig. 9E and F). Overall, our immunoprecipitation results clearly demonstrate the formation of a novel multiprotein complex consisting of at least BRE, CDC25A, USP7, and MERIT40 in response to DNA damage. The levels of this DNA damage-induced protein complex are increased in cells overexpressing BRE (Fig. 4e).

BRE contains three (+12 to +15 [PMLS]; +94 to +97 [PDPS]; and +201 to +204 [ALLS]) putative USP7-binding sequences (P/AxxS) that were previously identified in other USP7 interacting proteins[54]. We further tested if these sequences are important for binding to USP7 after irradiation. We generated different deletion mutants of BRE tagged with HA on their C-terminus (Fig. 4f upper panel). Immunoprecipitation with USP7 antibody pulled down all different deletion mutants of BRE except the one lacking all three USP7-binding sequences (FLΔ123, Fig. 4f). Next, we examined the region of USP7 responsible for binding to BRE. We generated two deletion mutants of USP7, either containing N-terminal TRAF (tumor necrosis factor (TNF) receptor-associated factor) domain or C-terminal region without TRAF domain (Supplementary Fig. 9G)[55]. Immunoprecipitation with BRE antibody shows that the interaction is mediated via N-terminal TRAF domain (Supplementary Fig. 9G).

We then tested if the disruption of the interaction between BRE and USP7 affects the known function of BRE in the stability of RAP80 or BRCC36 proteins[56]. We knocked down BRE in NIH3T3 cells and expressed either HA-BRE or HA- FLΔ123 BRE protein (human BRE cDNA) and examined RAP80 and BRCC36 protein levels (Supplementary Fig. 10). Expression of HA-BRE or HA- FLΔ123 BRE restores RAP80 and BRCC36 protein level after mouse BRE knockdown, indicating that HA- FLΔ123 BRE does not affect the key function of BRE (Supplementary Fig. 10). We also examined the effect of BRE FLΔ123 in MCF7 cells on CDC25A levels in response to IR. While BRE overexpression suppressed IR-induced CDC25A degradation, overexpression of BRE FLΔ123 failed to do that (Fig. 4g). This provides a strong evidence that BRE overexpression prevents IR-induced CDC25A degradation and its interaction with USP7 is essential for this.

**Survival of $Brca2^{KO/KO}$ mES cells by CDC25A overexpression.** Next, we overexpressed BRE defective in binding to USP7 (FLΔ123) in PL2F7 mES cells and selected the stable clones expressing the mutated BRE. We then examined its effect on IR-induced CDC25A stabilization. Similar to MCF7 cells, over-expression of BRE FLΔ123 failed to stabilize CDC25A in mES cells after irradiation (Fig. 5a). Further, we tested the ability of the mutant BRE to rescue the lethality of $Brca2^{KO/KO}$ cells. After

deletion of conditional $Brca2$ allele in those cells by $CRE$ expression, we failed to obtain any viable $Brca2^{KO/KO}$ cells (0/345 colonies screened) (Fig. 5b, upper panel). As control, when conditional $Brca2$ allele was deleted from BRE overexpressed cells, we obtained several $Brca2^{KO/KO}$ cells (14/276 colonies screened) (Fig. 5b, lower panel). These data support that the survival of $Brca2^{KO/KO}$ cells after BRE overexpression is due to its effect on CDC25A stabilization.

Next, we asked whether higher levels of CDC25A could rescue the lethality of $Brca2^{KO/KO}$ cells. To directly test this, we expressed $Flag$ epitope tagged $Cdc25A$ cDNA from $MSCV LTR$ in PL2F7 mES cells (Fig. 5c). We selected two independent clones in which, after IR, the level of CDC25A remained higher compared to irradiated control cells (Fig. 5c, lanes 4, 6). Further, when we deleted the conditional $Brca2$ allele in those cells by $CRE$ expression, we obtained viable $Brca2^{KO/KO}$ cells (Fig. 5d). These CDC25A overexpressing $Brca2^{KO/KO}$ cells ($Brca2^{KO/KO}$; $MSCV$-$Cdc25A$) showed defects in IR-induced RAD51 foci formation, increased chromosomal instability, and increased sensitivity to DNA-damaging agents indicating the impairment of $Brca2$ DSB repair function (Supplementary Fig. 11). Taken together, these data further support that BRE overexpression supports the viability of $Brca2^{KO/KO}$ cells by suppressing DNA damage-induced CDC25A degradation.

The effect of BRE on cell viability is independent of the DSB repair function of BRCA2. Therefore, we hypothesized that BRE overexpression should be able to rescue the cell lethality associated with loss of other DNA repair genes like $Brca1$, as well. To test this, we overexpressed HA tagged $BRE$ from $MSCV LTR$ into previously described[57] PL2F8 $Brca1$ conditional mES cells ($Brca1^{CKO/KO}$) (Supplementary Fig. 12A). Upon deletion of the conditional $Brca1$ allele, we obtained $Brca1^{KO/KO}$ clones in BRE overexpressing cells ($Brca1^{CKO/KO}$; $MSCV$-$BRE$) but none in the parental PL2F8 ($Brca1^{CKO/KO}$) cells (Supplementary Fig. 12B). This further supports that $BRE$ overexpression affects general cellular machinery rather than compensate for the loss of a specific function of BRCA2.

Further, we analyzed publicly available data sets for CDC25A expression in BRCA1/2 mutant tumors. We observed higher CDC25A mRNA expression in tumors carrying mutations in BRCA1/2 genes in multiple data sets (Fig. 5e, Supplementary Table 1). We next examined the correlation between BRE and CDC25A levels in BRCA2 mutant human breast tumors. We evaluated the expression of BRE in 15 breast tumors by immunohistochemistry. We found that 9/15 (60%) tumors from BRCA2 mutation carriers had higher BRE expression while six tumors showed relatively low levels (Fig. 6a). Interestingly, of the nine BRE high tumors, eight showed higher CDC25A expression

**Fig. 3** BRE overexpression affects IR-induced CDC25A degradation. **a** DNA synthesis in ES cells after different doses of IR. **b** Western blot analysis of CDC25A and Tyr15 phosphorylation of CDK1 (p-CDK1) at different times after exposing the cells to 6 Gy IR in $Brca2^{CKO/KO}$;$MSCV$-$BRE$ cells compared to $Brca2^{CKO/KO}$ cells. GAPDH was used as a loading control. Numbers below indicate the lane numbers. The quantification of CDC25A (middle panel) and p-CDK1 (lower panel) band intensity for each cell line is shown as histogram. Relative band intensities were calculated by dividing GAPDH-normalized CDC25A/p-CDK1 intensity at particular time point with GAPDH-normalized CDC25A/p-CDK1 intensity of corresponding untreated cells. **c** MCF7 cells with inducible $HA$-$BRE$ expression cassette were transfected with $CRE$ plasmid and subjected to 6 Gy IR after 48 h of transfection. Western blots of endogenous CDC25A in presence (lanes 4–6) and absence (lanes 1–3) of $HA$-$BRE$ expression at different times after irradiation is shown in the upper panel. Lower panel shows the quantification of CDC25A band intensity normalized against GAPDH. **d** Immunoblot showing CDC25A degradation after IR treatment in MCF7 cells transfected either with non-specific (NS) siRNA or two different siRNAs against BRE (#1 and #2). Histone H3 was used as a loading control. Relative CDC25A band intensities were calculated by dividing GAPDH-normalized CDC25A intensity at a particular time point with GAPDH-normalized CDC25A intensity of untreated (0 h IR) cells of corresponding siRNA treatment. **e** Western blot showing IR-induced CDC25A degradation in MCF7 cells with stably integrated CRE inducible $HA$-$BRE$ expression cassette after transfection either with non-specific (NS) siRNA or two different siRNAs against MERIT40 (#1 and #2) along with $CRE$-expressing plasmid. Non-specific (NS) siRNA transfection was used as control. Relative abundance of CDC25A at a particular time point was measured by comparing GAPDH-normalized band intensities at that point divided by GAPDH-normalized CDC25A intensity of untreated (0 h IR) cells of corresponding treatment. All histograms show the average of three independent experiments and error bars represent s.d. P values were calculated using paired two-tailed t-test and tabulated in Supplementary Table 2

(Fig. 6a). These results suggest a correlation between high CDC25A expression and higher BRE expression in BRCA-deficient breast tumors.

**BRE overexpression supports growth of BRCA2-deficient cells.** BRE overexpression is known to promote the growth of liver tumors[58]. Besides, high BRE expression correlates with poor

prognosis in non-radiotherapy treated breast cancer patients[59]. We therefore examined the effect of overexpression of BRE on the growth of *Brca2*-deficient cells in mice. We stably expressed HA-tagged BRE in KB2P1.21 mouse mammary tumor cells derived from $K14$-$Cre;Brca2^{F11/F11}$; $p53^{F2-10/F2-10}$ and then injected the cells into athymic nude mice and monitored tumor growth. *Brca2*-deficient BRE overexpressing cells (KB2P1.21 + BRE) showed accelerated tumor growth relative to cells without BRE

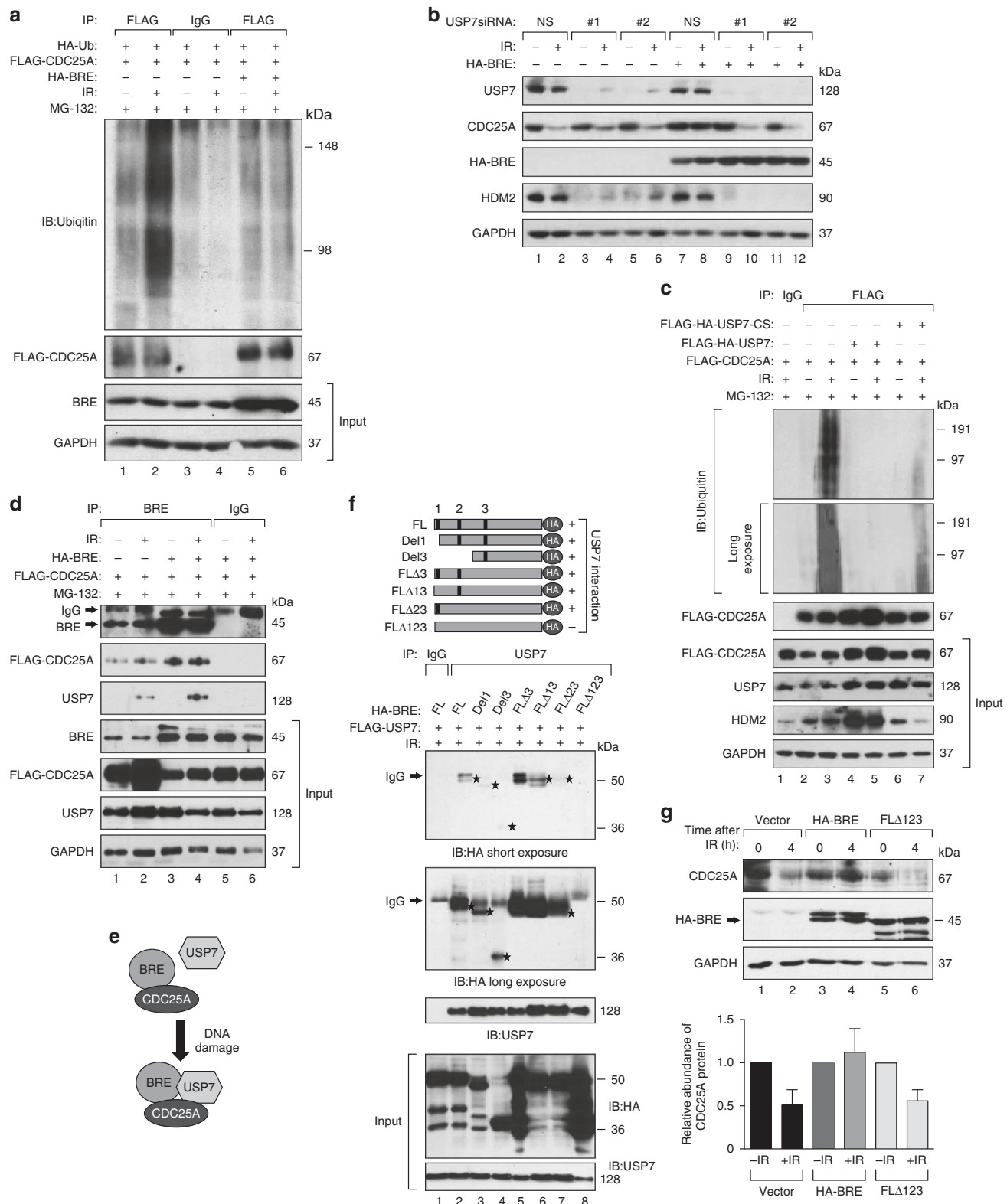

overexpression (Fig. 6b). We examined the level of BRE and CDC25A in these tumors and found that higher level of expression was maintained in tumors derived from BRE overexpressing cells (Fig. 6c, top panel). Also, the tumors that showed higher BRE expression exhibited high CDC25A expression (Fig. 6c middle panel). These results demonstrate that BRE overexpression promotes the growth of *Brca2*-deficient cells in vivo by regulating CDC25A.

## Discussion

The mechanism underlying the development of cancer due to loss of BRCA1 and BRCA2 has been explained by the 'caretaker and gatekeeper' hypothesis[60]. According to this hypothesis, BRCA proteins are considered to be 'caretakers' and loss of 'caretaker' function is essentially detrimental to cells. In response to loss of BRCA1/2, cells activate checkpoint mechanisms to prevent cell cycle progression and induce apoptosis in the presence of unrepaired DNA damage. Thus, the survival of BRCA-deficient cells either requires its cellular function complementation or loss of other genetic factors or the loss of a 'gatekeeper' function that can bypass the activated checkpoint mechanisms. The combination of those two events together can contribute to the development of cancer. One such combination is the loss of TRP53 along with the loss of BRCA1/2[13,61]. Here, we report another mechanism of survival of BRCA-deficient cells by BRE-mediated deregulation of CDC25A.

The regulation of CDC25A by BRE is dependent on USP7. We found that the interaction between CDC25A and USP7 is mediated by BRE. USP7 interacts with BRE via its N-terminal TRAF domain which either binds one or all three USP7 binding (P/AXXS) motifs of BRE. Presently, we do not understand the relevance of three USP7 binding domains in BRE. Presence of any one of these sites can be sufficient for CDC25A stabilization, we cannot rule out the possibility of synergistic effect of those sites on binding and protein stability.

In the presence of high BRE, the impairment of CDC25A protein regulation affects the cellular proliferation after BRCA2 loss. It is intriguing how BRE overexpression contributes to *Brca2*-null ES cell survival in the absence of IR. We propose that BRE overexpressing mES cells continue to divide even after the deletion of the conditional *Brca2* allele due to the presence of residual BRCA2. This is likely to result in sub-optimal BRCA2 activity that may lead to the accumulation of some DSB. We predict that this genomic instability may be sufficient to induce the formation of enough BRE-CDC25A-USP7 complex that results in CDC25A stability allowing cell cycle progression even after BRCA2 is depleted (Fig. 6d). The fact that the survival of $Brca2^{KO/KO}$ ES cells relies on multiple events explains why we

observed rescue of only 8–10% of HAT-resistant cells after *Brca2* deletion in the two independent BRE overexpressing clones (Fig. 1e). The effect of BRE overexpression on CDC25A is dependent on the presence of damaged DNA, and it is thus likely that BRE may therefore contribute to the tumorigenesis associated with mutation in other DNA repair genes such as *PALB2*, *RAD51C*, etc.

Higher CDC25A expression has been reported in breast and other carcinoma and is associated with poor prognosis[62,63]. Previously, the deregulation of CDC25A protein has been correlated with increased expression of ubiquitin hydrolase DUB3 in breast cancer cell lines or dysfunction of β-TRCP in lung cancer[46,64]. However, studies using transgenic mouse models have shown that when its overexpression is combined with other oncogenic mutations like HRAS or C-NEU, it accelerates tumor growth[65]. These findings support the notion that when combined with high genomic instability[66,67], CDC25A may induce tumorigenesis in BRCA1/2-deficient cells.

A recent study found the combination of replicative stress induced by ATR inhibitors and premature mitotic entry by high CDC25A expression to be toxic to cells[68]. Therefore, it is confounding how high CDC25A expression helps in the survival of BRCA2-deficient cells. While the precise mechanism remains unknown, we propose that the presence of high CDC25A in the absence of BRCA2 can promote mitotic entry of such cells. This can exert mitotic catastrophe. It is possible that some cells may accumulate secondary mutations due to their unstable genome and these mutations can then support cell viability. This is supported by the low number of rescued $Brca2^{KO/KO}$ cells obtained by BRE or CDC25A overexpression. Similarly, when we examined the effect of BRE overexpression on the growth of MCF7 cells after BRCA2 knockdown, cell proliferation was found to be comparable to the wild-type cells at initial time points (Fig. 1f, days 1–5). At later time points (Fig. 1f, after day 5), the cell proliferation was significantly reduced compared to the wild-type cells. This suggests that although BRE overexpression contributes to resistance to BRCA2-loss-induced growth arrest, not all cells survive or proliferate to the same extent.

In conclusion, we identified a novel mechanism of survival of BRCA2-deficient cells by CDC25A, which is frequently upregulated in tumors from *BRCA1/2* mutation carriers. Furthermore, we have also uncovered a novel multiprotein complex containing BRE-USP7-CDC25A that has revealed a new mechanism of CDC25A regulation. Considering that CDC25A can act as a biomarker for ATR inhibitor sensitivity[68] and the increased expression of CDC25A in BRCA1/2 tumors (Fig. 5e), it will be intriguing to test such tumors for their sensitivity to ATR inhibitors. Also, combining ATR inhibitors with PARP inhibitors can be efficacious for treating BRCA1/2 tumors.

**Fig. 4** DNA damage-induced BRE-USP7 interaction stabilizes CDC25A. **a** Ubiquitylation of FLAG-CDC25A in cells harvested 4 h after 6 Gy IR in absence (lanes 1–4) or presence (lanes 5, 6) of exogenous BRE. CDC25A was immuno-precipitated using anti-FLAG antibody (lanes 1, 2, 5, 6), IgG was used as control (lanes 3, 4). **b** MCF7 cells stably expressing CRE-inducible *HA-BRE* were transfected with either non-specific (NS, lanes 1–2 and 7–8) or two independent USP7-specific (USP7 #1, lanes 3, 4, 9, 10 and USP7 #2, lanes 5, 6, 11, 12) siRNAs along with or without plasmids expressing CRE and irradiated with 6 Gy IR. Western blot shows the abundance of endogenous CDC25A in non-IR cells and 4 h after irradiation. HDM2 was used as a control for USP7 knockdown. **c** FLAG-CDC25A ubiquitylation in cells harvested 4 h after 6 Gy IR in absence (lanes 1–3) or in presence of exogenous USP7 (lanes 4, 5) or catalytic inactive USP7 (USP7-CS; lanes 6, 7). Anti-FLAG antibody (lanes 2–7) was used to pull down CDC25A, IgG was used as control (lane 1). HDM2 was used as a control for USP7 activity. **d** Co-immunoprecipitation of FLAG-CDC25A, BRE, and endogenous USP7 from MCF7 cells using anti-BRE antibody (lanes 1–4) and IgG antibody (lanes 5–6); 5 μM MG132 was added after irradiation to increase CDC25A level and harvested after 2 h. **e** Model depicting the interaction of BRE, CDC25A, and USP7. In presence of DNA breaks, BRE interacts with USP7 and then BRE, USP7, and CDC25A complex is formed. For simplicity, MERIT40 is not included in the model. **f** Co-immunoprecipitation of USP7 and different deletion mutants of BRE tagged with HA-epitope from irradiated HEK293 cells using USP7 antibody. Top panel shows the schematic representation of different deletion mutants of BRE. Numbers in the top panel mark different USP7 binding motifs. Stars indicate different HA-BRE proteins. **g** Western blots showing IR-induced degradation of CDC25A after expressing HA-BRE or HA-BRE mutant defective in USP7 binding (FLΔ123). GAPDH was used as a loading control for all blots. Histogram represents mean ± s.d. values of relative band intensities. *P* values are calculated using the paired two-tailed *t*-test and presented in Supplementary Table 2

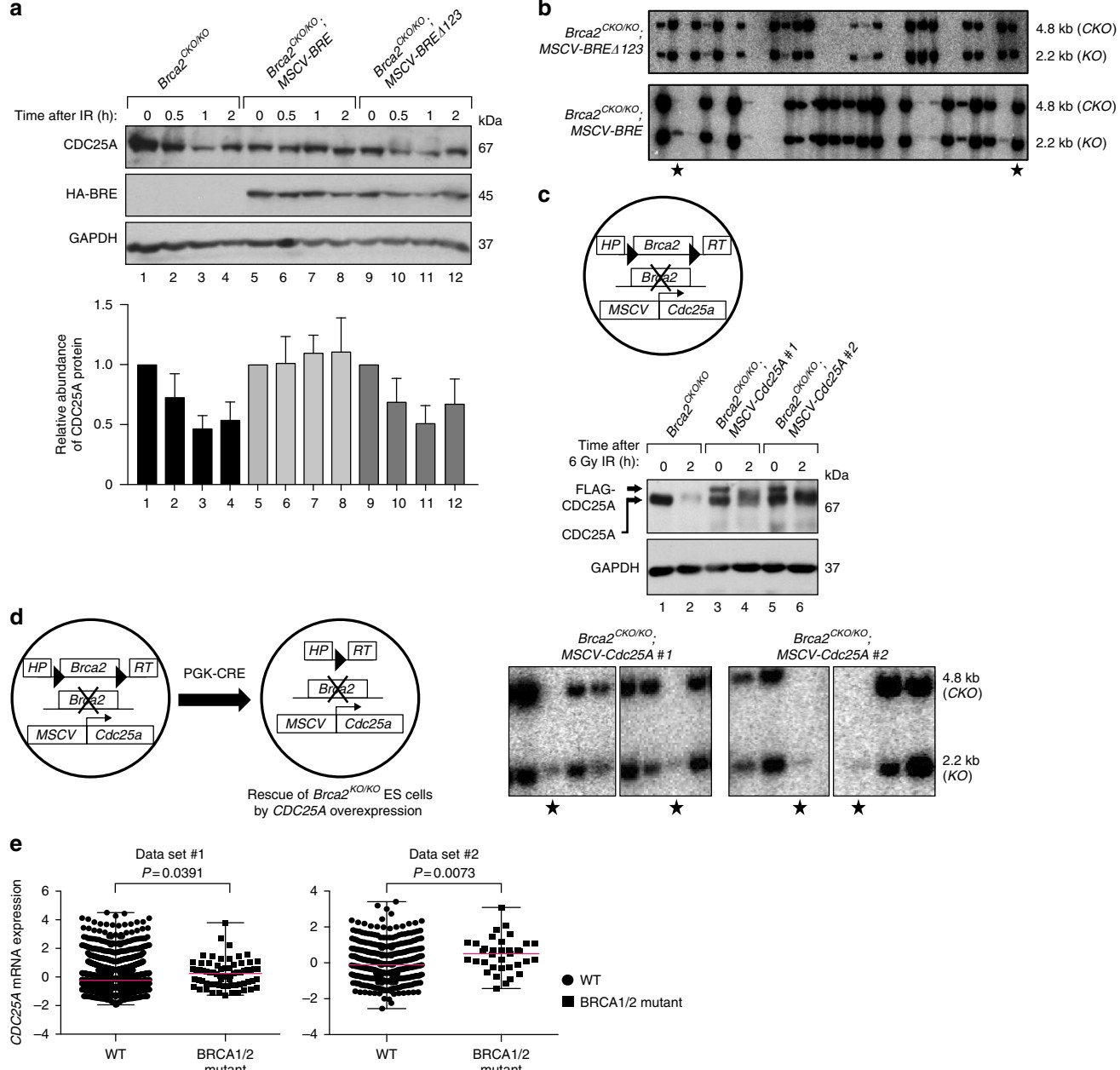

**Fig. 5** CDC25A overexpression can rescue ES cell lethality of $Brca2^{KO/KO}$. **a** Immunoblot showing IR-induced degradation of CDC25A in PL2F7 mES cells expressing either HA-BRE or HA-BRE mutant defective in USP7 binding (FLΔ123). GAPDH was used as control. The histogram below shows the average relative abundance of CDC25A at particular time points from three independent experiments. All intensities were normalized against corresponding GAPDH band intensities. Supplementary Table 2 shows the P values. **b** Representative Southern blot analysis showing the identification of $Brca2^{KO/KO}$ cells (marked with solid stars) after CRE-mediated deletion of conditional allele in indicated cell lines. Upper band: conditional allele (CKO); lower band: knock-out allele (KO). **c** Western blot showing CDC25A abundance in $Brca2^{CKO/KO}$ and $Brca2^{CKO/KO}$; MSCV-Cdc25a ES cells before (lanes 1, 3, 5) and 2 h after 6 Gy IR (lanes 2, 4, 6). Top panel represents the scheme of expression of FLAG-CDC25A from MSCV LTR. GAPDH was used as a loading control. **d** Southern blot analysis of HAT-resistant ES cell colonies after CRE-mediated deletion of conditional allele in $Brca2^{CKO/KO}$; MSCV-Cdc25a ES cells to identify $Brca2^{KO/KO}$ clones (marked with solid stars), upper band: conditional allele (CKO); lower band: knock-out allele (KO). **e** CDC25A mRNA expression in two different breast cancer data sets from The Cancer Genome Atlas (TCGA). Left panel represents the data set of METABRIC breast cancer with 2433 tumors and right panel shows the breast invasive carcinoma data set of 825 tumors. Dots or squares in graphs represent individual tumors. Middle line, median; whiskers, minimal and maximum values. Significance was calculated using unpaired t-test with Welch's corrections

## Methods

**Plasmids, antibodies, and reagents**. MSCV-Cre plasmid was a kind gift from Dr. Dan Littman. HA epitope was tagged to the C-terminus of human *BRE* cDNA and was cloned in to MSCVneo vector (Clontech #634401). To generate *MSCV-Cdc25a* expression vector, mouse *Cdc25A* cDNA was N-terminally fused with FLAG epitope and cloned into MSCVneo vector (Clontech). For inducible expression of BRE, HA-tagged human *BRE* cDNA was cloned into vector pCCALL2-IRES-

GFP[69]. The pCDNA3.1+ (Invitrogen) vector was used for transient expression of *FLAG-Cdc25A, HA-BRE*, and *FLAG-DUB3*. For the expression of BRCC36, Ubiquitin, USP7, catalytic inactive USP7, β-TRCP3, RAP80, MERIT40, and BRCA1 pOZ-N-FH BRCC36 (Addgene #27496) FLAG-HA-BRCC3 (Addgene #22540), HA-Ubiquitin (Addgene #18712), pQHA-USP7 WT puroR (Addgene #46753), pCI-neo-FLAG-HAUSP (Addgene #16655), pQHA-USP7 CS puroR (Addgene #46754), pTRCP3-Myc (Addgene #25902), pOZ-N-FH RAP80 (Addgene #27501),

pOZ-N-FH MERIT40 (Addgene # 27498) and pBABEpuro HA Brca1 (Addgene #14999) were used, respectively. The antibodies used for co-immunoprecipitation, immunofluorescence, and western blots were anti-BRE (C-16: sc-48847 and sc-376453; Santa Cruz Biotech, 1:500 dilution for western blots), anti-BRCC36 (ab62075; Abcam and sc131122; Santa Cruz Biotech, 1:1000 dilution for western blots), anti-CDC25A (F-6: sc-7389; Santa Cruz Biotech, 1:200 dilution), anti-FLAG

(F7425 and F3165; Sigma-Aldrich, 1:2000 dilution for western blot), anti-HA (12013819001; Roche, 1:2000 dilution), anti-USP7 (H-200: sc-30164 and H12: sc-137008; Santa Cruz Biotech and ab4080; Abcam, 1:1000 dilution for western blot), anti-β TRCP (ab118006; Abcam, 1:500 dilution), anti-DUB3 (ab174914; Abcam and H00377630-M01; Abnova, 1:1000 dilution for western blot), anti-RAP80 (A300-764A-M; Bethyl laboratories, 1:500 dilution for western blot), K-48 linkage

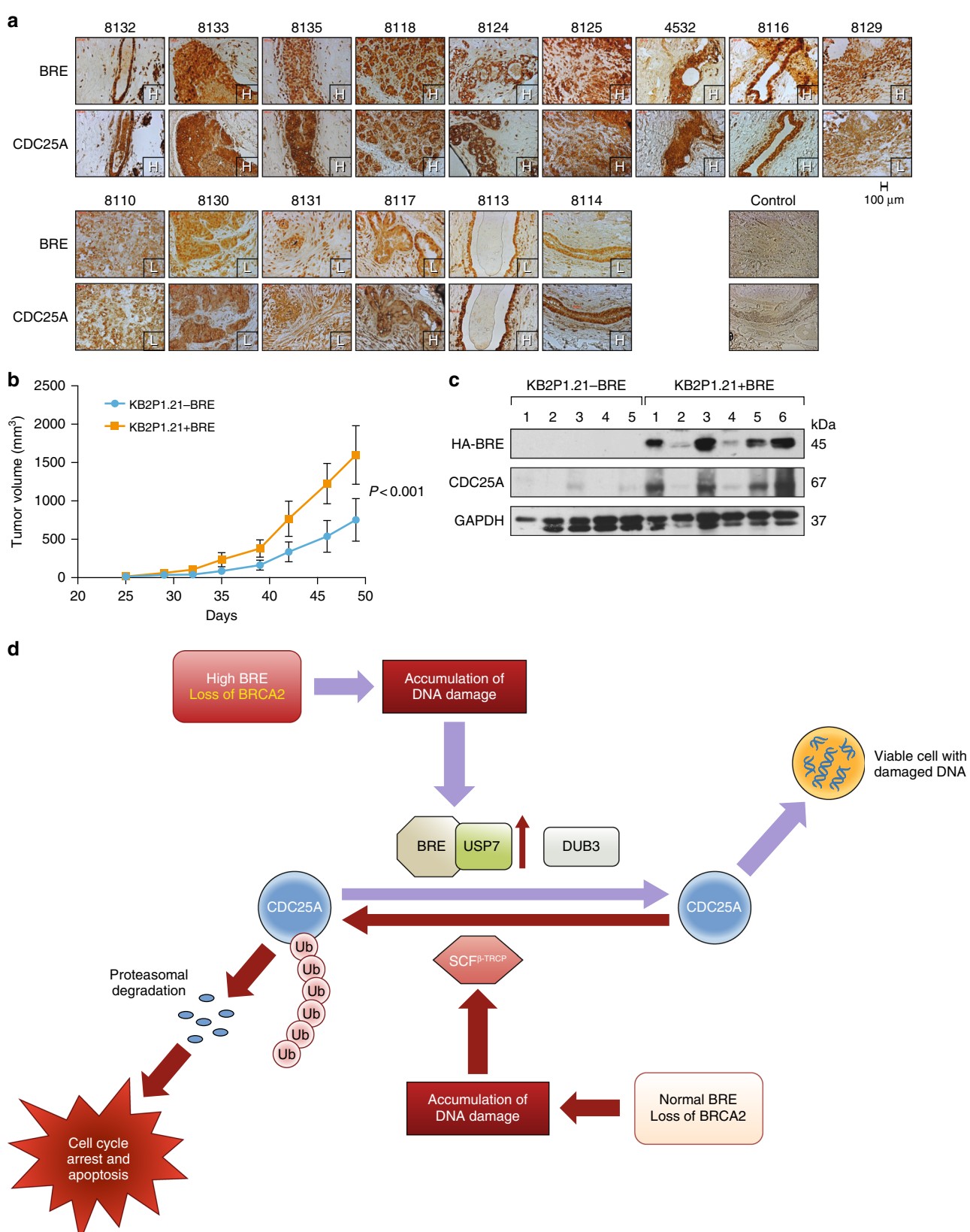

specific antibody (#4289S; Cell Signaling, 1:500 dilution), K-63 linkage specific antibody (#5621S; Cell Signaling, 1:500 dilution), anti-histone H3 (#9715; Cell Signaling, 1:5000 dilution), anti-HDM2 (sc-56154; Santa Cruz Biotech, 1:500 dilution), anti GAPDH (2118S; Cell Signaling, 1:40,000 dilution), anti-ACTIN (C2: sc-8432; Santa Cruz Biotech, 1:40,000 dilution), anti-phosphorylated CHK1 (12302; Cell Signaling, 1:500 dilution), anti-phosphorylated CDK1 (#9111; Cell Signaling, 1:1000 dilution), anti-p53 (#2524; Cell Signaling, 1:5000 dilution), anti-RAD51 (PC130; Calbiochem, 1:2000 for western and 1:250 for Immuno-fluorescence), anti-γH2AX (05-636; Upstate), anti-ubiquitin (P4D1: sc-8017; Santa Cruz Biotech, 1:1000 dilution), anti-Myc (631206, Clontech and 2272S; Cell Signaling, 1:1000 dilution), anti-MERIT40 (#12711; Cell Signaling, 1:1000 dilution), and anti-BRCA2 (A303-434A-T; Bethyl Laboratories, 1:2000 dilution).

USP7 inhibitor P5091(#S7132) was obtained from Selleckchem and used at 10 μM concentration.

**Cell culture and transfection**. All mESCs were cultured on top of mitotically inactive SNL feeder cells in M15 media (knockout DMEM media (Life Technologies) supplemented with 15% fetal bovine serum (FBS; Life Technologies), 0.00072% beta-mercaptoethanol, 100 U ml$^{-1}$ penicillin, 100 μg ml$^{-1}$ streptomycin, and 0.292 mg ml$^{-1}$ L-glutamine) at 37 °C, 5% CO$_2$. PL2F7 and PL2F8 cells are derivatives from AB2.2 mouse embryonic stem cell line[6,57].

The human breast cancer cell lines MCF7 (obtained from American Type Culture Collection (ATCC), Rockville, MD) and mouse NIH 3T3 (obtained from American Type Culture Collection (ATCC), Rockville, MD) were cultured in DMEM (Thermo Fisher Scientific); human prostate cancer cell line PC3 (obtained from National Cancer Institute cell line repository, Frederick, MD) was cultured in F-12K medium (Thermo Fisher Scientific), human colon cancer cell line SW620 (obtained from National Cancer Institute cell line repository, Frederick, MD) was cultured in Leibovitz's L-15 medium (Thermo Fisher Scientific), and human breast cancer cell line ZR75-1 (obtained from American Type Culture Collection (ATCC), Rockville, MD) was cultured in RPMI1640 (Thermo Fisher Scientific). All media were supplemented with 10% FBS and antibiotics. Mouse mammary tumor cell lines (KB2P1.21) were cultured at 37 °C, 5% CO$_2$, 3% O$_2$ in DMEM/F-12 (Life Technologies) supplemented with 10% FBS, antibiotics. ES cells were cultured on mitotically inactivated feeder cells as described previously[6].

Human and mouse cancer cell lines were transfected with indicated plasmids using Lipofectamine 2000 (Life Technologies) according to manufacturer's protocol. To generate MCF7 cell lines with inducible expression of HA-BRE, MCF7 cells were transfected with plasmid expressing inducible HA-BRE and selected on 700 μg ml$^{-1}$ G418 (Gibco) for 2 weeks. Individual colonies were expanded and analyzed for inducible expression of HA-BRE. To induce the expression of HA-BRE, cells were transfected with plasmids expressing *CRE* from *Pgk* promoter.

All cell lines were tested routinely for mycoplasma contamination.

**Knockdown of *BRCA2*, *BRE*, *MERIT40*, and *USP7***. Lentiviral shRNA vector against human *BRCA2* (TRCN0000010306 and TRCN0000009825) and control shRNA (SHC202) were purchased from Sigma (MISSION shRNA). HEK293T cells were used for packaging. HEK293T cells were co-transfected with shRNA vector, pRSV-Rev, pMDLg-pRRE, and pHCMVG. Packaging vectors were kind gifts from Dr. Steven Hou (NCI-Frederick, NIH). After 72 h transfection, the supernatant was collected and 0.45 μm filtered before being used for infecting either MCF7Neo (G418 resistant MCF7 cells) or MCF7BRE (BRE overexpressing) cells. After 48 h infection, cells were subjected to 2 μg ml$^{-1}$ puromycin selection for 2 days and puromycicn-resistant cells were pooled and used further.

For knockdown of *BRE*, siRNAs (SR306396A, SR306396B, and SR306396C) against human *BRE* were purchased from Origene along with control (SR30004). Mouse *BRE* siRNAs (#s99035 and #s99036), USP7 siRNAs (s15439 and s15440), MERIT 40 siRNAs (#124756 and #20769 of cat # AM16708), and control siRNA (cat# 4390843) were obtained from Ambion. Transfection of siRNA was carried out using Lipofectamine RNAiMAX transfection reagent (Life Technologies).

**Growth analysis of MCF7 cells**. A total of 200,000 cells were seeded in triplicate for each cell line in a 6-cm dish. At indicated time points, cells were trypsinized, harvested, and counted using Beckman coulter counter. Fold growth was estimated by dividing the cell number at a particular time point by the cell number after 24 h of seeding. Each plate was counted thrice and the average was taken. For each time point, an average of three independent plates was plotted.

**DNA fiber assay**. After pulsing sequentially by 8 μg ml$^{-1}$ CldU for 15 min followed by 90 μg ml$^{-1}$ IdU for 15 min, the cells were treated with 4 mM HU for 3 h and were resuspended in PBS. Approximately $3 \times 10^5$ cells in 3 μl volume were mixed with 7 μl lysis buffer (200 mM Tris-HCl (pH 7.4), 50 mM EDTA, 0.5% SDS) on glass slides and incubated at room temperature for 8 min before DNA fiber was spread. Fibers were then fixed in methanol and acetic acid (3:1) at 4 °C overnight, rehydrated by PBS followed by denaturation in 2.5 M HCl for 1 h. After rinsing away HCl by PBS, slides were then blocked in PBS with 5% bovine serum albumin (BSA) for 1 h and incubated with anti-BrdU antibody (mouse, #347580, Becton Dickinson, 1:100 dilution) and anti-BrdU antibody (rat, ab6326, Abcam, 1:500 dilution) at 4 °C overnight. Next day, slides were rinsed with PBS with 0.1% Tween-20 (PBST) and incubated with AlexaFluo488-conjugated anti-mouse IgG secondary antibody and AlexaFluo594-conjugated anti-rat IgG secondary antibody for 1 h at room temperature. After rinsing with PBST, the slides were mounted by mounting media (Prolong Gold, Invitrogen). the images were captured in Zeiss Axio Imager Z1 microscope and the fiber length was measured by ImageJ software (NIH).

**Co-immunoprecipitation**. After 36–48 h of transfection, the cells were irradiated with 6 Gy γ-radiation and treated with 5 μM MG132 (Sigma Aldrich) for 2 h where indicated. Total proteins were extracted in IP buffer (20 mM HEPES (pH 7.5), 100 mM NaCl, 1 mM EDTA, 1 mM EGTA, 1 mM NaF, 1 mM DTT, 0.1% Triton X-100, 1 mM PMSF, protease inhibitor cocktail (Roche), phosphatase inhibitor cocktail (Roche)). Immunoprecipitation was carried out by incubating pre-cleared lysates with protein-G agarose beads (Roche) and indicated antibodies at 3–5 μg/mg of protein lysates for 4–6 h. Then the beads were washed in lysis buffer for 4–5 times and the immunoprecipitated complex was separated either in Tris-Glycine gel or 4–12% Bis-Tris gels by electrophoresis and detected with appropriate antibodies.

For BRE deletion constructs, desired fragments were either amplified using PCR or synthesized using GeneArt gene synthesis (Thermo Fisher) and cloned into pcDNA3.1(+) vector DNA. Del1 has 2–18 aa and Del3 has 2–147 aa deletions. Deletion of motifs 1, 2, and 3 represents deletion of aa 12–15, aa 94–97, and aa 201–204, respectively. All constructs of BRE deletions or full length were tagged with HA-epitope at C-terminus. For USP7 deletions, all constructs were tagged with MYC-epitope at their N-terminus and cloned into pcDNA3.1(+) vector DNA. Del-C contains deletion of C-terminal amino acids 211–1102 whereas Del-N contains deletion of N-terminal amino acids 2–210. After cloning, the deletions were confirmed by sequencing. The desired plasmids were then transfected into HEK293 (ATCC) cells. After 36 h of transfection, the cells were irradiated with 6 Gy γ-radiation. Proteins were isolated 2 h after irradiation and immunoprecipitation was carried out.

**In vivo ubiquitylation assay**. Ubiquitylation assay was performed as described previously[70]. Briefly, 36h after transfection of plasmids, the cells were irradiated and treated with MG132 for 4 h. Total proteins were isolated in lysis buffer (30 mm Tris, pH. 8, 75 mm NaCl, 10% glycerol, and 1% Triton X-100) and equal amounts of total protein were immunoprecipitated with FLAG antibody (3 μg/500 μg of lysate protein; Sigma Aldrich). Amount of ubiquitylation was measured by detecting the extent of ubiquitylation with an anti-ubiquitin (P4D1; 1:500; Santa Cruz Biotechnology) antibody that detects both poly- and mono-ubiquitylated proteins using western blot analysis. For detecting K-63 and K-48 linked poly-ubiquitylation, same blot was stripped using Restore™ Western Blot Stripping Buffer (Thermo Fisher Scientific) and used sequentially with K-63 and K-48 specific antibodies.

**CDC25A mRNA expression levels in human tumor data sets**. Two independent breast cancer data sets were considered from The Cancer Genome Atlas[71,72]. CDC25A mRNA expression was considered in samples with and without *BRCA1/2* mutations. Statistical analysis was done using unpaired *t*-test with Welch's correction using Graphpad Prism software.

**Immunohistochemistry and tumor tissue array**. Fifteen BRCA2 mutant breast tumor sections (from Lombardi Cancer Center, Georgetown University, Washington DC) were used for BRE and CDC25A expression analysis. Slides containing tumor samples were deparaffinized, hydrated, and boiled in citrate buffer for 20 min for antigen exposure. After blocking, the slides were probed either

**Fig. 6** Physiological relevance of BRE overexpression. **a** Immunohistological staining of 15 BRCA2-deficient tumors using BRE and CDC25A antibodies. Numbers represent different tumor numbers. Control represents staining with control IgG. H: high expression, L: low expression. **b** Allograft tumor growth of KB2P1.21 (*Brca2* negative) cells with and without stable expression of HA-BRE. Values are represented as mean ± s.e.m. *P* values were calculated using ANCOVA test. **c** Western blot showing the expression of HA-BRE and CDC25A in tumors from allograft experiment. GAPDH was used as control. **d** Proposed model explaining the survival of BRE overexpressed BRCA2-deficient cells. CDC25A protein stabilization is regulated by the balanced action of ubiquitylation and deubiquitylation mediated by β-TRCP and DUB3, respectively. In cells with normal BRE expression level, loss of BRCA2 induces DNA damage that leads to β-TRCP-mediated degradation of CDC25A and cells undergo cell cycle arrest and apoptosis. BRCA2 loss in BRE overexpressing cells results an increase in BRE/USP7/CDC25A complex that perturbs the balance of CDC25A ubiquitylation/deubiquitylation and leads to an increase in CDC25A even in presence of DNA damage. This elevated level of CDC25A leads to the survival of cells with damaged DNA

with anti-BRE antibody (GTX20991; Genetex, 1:200 dilution) or with anti-CDC25A antibody (sc-97; Santa Cruz Biotech, 1:400 dilution). For control, one slide was stained with normal IgG (Santa Cruz Biotech). The signal was detected by Elite ABC HRP detection kit (Vectastain). Microscopic observation was performed using the Axioplan2 upright microscope (Zeiss). Intensities of staining were measured using Image J software. Staining intensities with values >0.45 (arbitrary unit) were considered as high expression and lower than 0.45 were considered as low expression.

**Allograft assay**. Mouse tumor cell lines were harvested, counted, and diluted to a concentration of $2 \times 10^6$ cells/200 μl of L15 medium and injected sub-cutaneously into the flanks of 6-week-old nude mice. Six mice were used in each group. No selection criteria was used for selecting the mice for each group. Tumor growth was measured twice on every week blindly. Tumor volume (in $mm^3$) was calculated as a product of $2 \times$ length $\times$ width. Mice were maintained according to the procedures outlined in the Guide for the Care and Use of Laboratory Animals, under an approved Animal Care and Use Committee (ACUC) protocol.

**Statistics**. All data are expressed as means ± s.d., if not indicated. Statistics were performed using data from three independent experiments. For relative quantification of protein band intensity, loading control normalized intensity at a particular time point was divided by loading control normalized intensity of corresponding untreated cells. Differences between two groups were compared using a two-tailed Student's t-test (Prism). The data are normally distributed and $P < 0.05$ was considered statistically significant. For the allograft experiment, the difference between the two groups (BRE overexpressed vs. non-overexpressed) was measured by analysis of variance ANCOVA using Prism software. No statistical methods or criteria were used to estimate sample size or to include or exclude samples. All group allocations during the experiments were blinded by the investigators unless otherwise specified.

Additional methods are described in the Supplementary Methods section.

**Data availability**. Uncropped blots are presented in Supplementary Fig. 13. All relevant data are available from the authors on request.

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

## Acknowledgements

We thank members of our laboratory for helpful discussions and suggestions. We thank Dr. Michael Kuehn for help with MSCV transduction in ES cells and Drs. Jairaj Acharya, Ira Daar, Debananda Pati, Xia Ding, Suzanne Hartford, and Andre Nussenzweig for their comments and critical review of the manuscript. We thank Dr. Dan Littman (New York University, School of Medicine) for MSCV-Cre construct, Dr. Corrinne Lobe (Sunny-brook Health Science Center, Toronto, Ontario, Canada) for PCALL2-IRES-GFP plasmid, Drs. Jos Jonkers and Peter Bouwman (Netherland Cancer Institute) for mammary tumor cell lines. We thank Dr. Sudhirkumar Yanpallewar for help with immunohis-tochemistry and Dr. Erkan Kiris for help with in vivo ubiquitylation assays. We also thank Allen Kane (SAIC-Frederick, Inc., Scientific Publications, Graphics & Media Department) for help with the illustrations. This research was sponsored by the Intramural Research Program, Center for Cancer Research, National Cancer Institute, US National Institutes of Health.

## Author contributions

K.B. designed and performed the experiments, analyzed the data, and wrote the paper. S. P., A.Y., V.S., A.B. and S.C. designed and performed the experiments, analyzed the data. B.K.M., S.B. and S.L.N. provided technical help. S.K.S. conceived the study, analyzed the data, and wrote the paper.

## Additional information

**Competing interests:** The authors declare no competing financial interests.

