## [Peer Review File · Nature Communications]

Reviewers' comments:

Reviewer #1 (Remarks to the Author):

The manuscript entitled "BRE/BRCC45 Regulates CDC25A Stability by Recruiting USP7 in Response to DNA Damage" by Biswas et al describes the identification and characterization of the novel role of BRE, CDC25A and USP7 in the DNA damage response that result in survival of cells lacking BRCA2. One of the main players in genome integrity is BRCA2 which functions in homologous repair by interacting with BRCA1 and PALB1, integral components of the RAD51 complex. Deletion or mutation of BRCA2 leads to oncogenesis as the DNA damage response is disrupted. The results presented in this manuscript strongly support the newly identified function of the BRE:CDC25A:USP7 complex in the DNA damage response. The identification of the mechanism by which BRE overexpression rescued the BRCA2KO/KO cells is very novel and significant as it provides a plausible route for oncogenesis.

This is a well carried out study that adds an important component to the DNA damage response mechanism. Using their insertional mutagenesis screen, the authors identified BRE as an important mediator of BRCA2KO/KO cell survival. This was followed up by the characterization of the function of BRE/BRCC45 and the identification of USP7 as a major regulator of CDC25A stability.

Major points.

There are no error bars or statistics in the blots of Figures 3, 4, 5 or supplementary. In addition, there is absolutely no mention in the text how many times these experiments were repeated. The authors should repeat these experiments at least 3 times and provide the necessary statistics for these blots.

Several of the blots are overexposed (Fig 3C, 4A, 6C) or otherwise dirty (CDC25A in Fig 3E). Also there are some inconsistencies with loading seen by GAPDH.

In Fig 4A can you show that its K48-linked polyubiquitination using K48-linked specific antibody?? Can you also show that there isn't any K63-linked polyubiquitination using the K63-linked specific antibody?

Explain this sentence further "Involvement of BRCC36 in selective cleavage of K63-linked polyubiquitin further explains its inability to participate in BRE-mediated CDC25A deregulation in response to DNA damage."

Does the addition of USP7 catalyze deubiquitination of ubiquitinated CDC25A? This should be assessed and added to Fig 4A. What about catalytically inactive USP7?

Figs 4B and 4C why do the levels of USP7 fluctuate (lanes 1-4)?

Describe sequence details of the BRE/BRCC45 USP7 binding motifs.

Which USP7 domain is mediating interaction? This should be demonstrated using the individual domains. Typically substrate proteins interact with the USP7-TRAF domain. In this case if BRE/BRCC45 contains the binding motif, how does USP7 catalyze deubiquitination of CDC25A? Describe the mechanistic details.

There is no discussion as to why there are 3 USP7 interaction sites within BRE/BRCC45. Are all required for stabilization of CDC25A?

Does CDC25A interact directly with USP7 in the absence of BRE/BRCC45?

What other known BRE/BRCC45 interacting proteins are components of this novel complex? Does BRCA1 still interact with BRE/BRCC45 when its in the BRE:USP7: CDC25A complex?

The absence of USP7 in the co-IP in Supplementary Fig 7C needs further comment from the authors. It looks like the BRE/BRCC45 IPed levels are similar in lanes 2 and 4 so why is USP7 missing from lane 2.

Are the same effects on the levels of CDC25A seen with a USP7 inhibitor? What about a USP7 active site mutant?

What happens to levels of CDC25A with overexpressed USP7?

Is BRE (FL Δ 123) still active in the BRISC or Rap80 complexes?

It would be nice to see a positive control in the siUSP7 studies. Do the levels of known USP7 substrates also decrease?

Minor points

There are several instances of very short sentences (for example page 4 line 70) which should be expanded or incorporated into the previous sentences.

Need a better, more comprehensive introduction on the functions of BRE/BRCC45 that what is currently found on page 4. In addition, these functions should be incorporated into the discussion.

Reviewer #2 (Remarks to the Author):

This is a review of NCOMMS-17-12698-T. In many ways, this is a two-part study. In the middle part, a novel function of the BRE protein in promoting CDC25A stability after DNA damage is described, and a strong correlation with BRE association with the deubiquitinase USP7, and USP7-mediated ubiquitination of CDC25A are shown. These findings are significant, because CDC25A is a key regulator of cell cycle progression and hyper-stabilization could favor tumorigenesis. These findings are novel and thorough, although the RNAi experiments are not as convincing as the overexpression experiments (Major point 1). The other part of the study reflects their interest in BRE in the first place - through an insertional screen, and followed up by overexpression experiments, elevated BRE levels enabled the isolation of BRCA2^{-/-} mouse ES clones. These clones are characterized as showing genome instability phenotypes consistent with BRCA2 loss, as well as an intra-S-phase checkpoint defect, which lead to the studies of CDC25A. Clinical relevance of such findings are supported by IHC analysis of BRE & CDC25A in breast cancers, as well as a xenograft experiment with a BRCA2-deficient tumor line (using BRE overexpression). These data are compelling, but the question, as mentioned in the Discussion, is why are these mES clones alive? The hypothesis in the Discussion is that these elevated BRE/CDC25A clones that survive without BRCA2 (or BRCA1) are nevertheless rare, and that perhaps secondary mutations / rearrangements were required for these clones to grow out. Accordingly, the degree of rescue that is supported by BRE or CDC25A overexpression per se is not well described (Major Point 2). Apart from these concerns, I found this to be a compelling study with novel insight into CDC25A regulation via BRE and USP7, and raises new ideas about the etiology of tumors with genome instability.

Major points

1. A more thorough time course with one cell line for BRE knockdown should be shown (e.g. every 30 min after IR for 4 hr for the lines in supplemental, or every 1hr for the line in the main figure). Given the marked changes between early time points after IR and a later time point, I found the effects of BRE depletion to not be entirely convincing.

2. The degree of rescue that is supported by BRE or CDC25A overexpression per se is not well described, apart from the fact that clones could be isolated. I found this to be a weakness of the study. I recognize that this is somewhat difficult, given the incomplete penetrance of CRE activation. However, could an inducible shRNA to BRCA2 (or BRCA1) be used to monitor cell proliferation over time in BRE and CDC25A overexpressed cells, either with the mouse ES or MCF7 models, as in this study (PMID: 23337117)? Namely, is there a resistance to BRCA depletion at early divisions with BRE and/or CDC25A overexpression, or do most of the cells die in the first few days, but then a resistant set of clones grow out. This is an important distinction, which relates to the paradox described in the Discussion regarding the ATR-inhibition work.

Minor point

1. mis-spelled PARP at the end.

RESPONSE TO THE REVIEWERS' COMMENTS

We would like to thank the reviewers for their overall very positive response and for their insightful comments and suggestions. We have performed several new experiments and made necessary changes in the manuscript to address their concerns. These changes have improved the overall quality of the manuscript.

Our point-by-point response to the major and minor comments is described below:

Reviewer #1

There are no error bars or statistics in the blots of Figures 3, 4, 5 or supplementary. In addition, there is absolutely no mention in the text how many times these experiments were repeated. The authors should repeat these experiments at least 3 times and provide the necessary statistics for these blots.

As suggested by the reviewer, we have now plotted the mean relative intensities with error bars and performed statistical analysis of all Western blots where we have quantified the band intensities (Figures 3B-E, 4G, 5A and Supplementary Figures 3, 4 and 5). In every case, the experiments were performed at least three times. We have included this information in the figure legends.

We have removed the Phospho-CDK1 blots and corresponding histograms from Supplementary Figure 3 because the phospho-CDK1 blot shown in Supplementary Figure 3A was performed only once in *Brca2^{KO/KO} Clone 3d*. We are unable to perform this experiment now due to the loss of BRE overexpression in the later passages of this clone. Levels of phospho-CDK1 were analyzed in *Brca2^{CKO/KO};MSCV-BRE* cell lines three times and the data is incorporated into Figure 3B. We also analyzed phospho-CDK1 levels in *Brca2^{KO/KO}; MSCV-BRE* cell lines three times but for consistency, the data was not included in Supplementary Figure 3B. If the reviewer thinks that the phospho-CDK1 data should be included in Supplementary Figure 3, we can include the blots without statistical analysis.

Several of the blots are overexposed (Fig 3C, 4A, 6C) or otherwise dirty (CDC25A in Fig 3E). Also there are some inconsistencies with loading seen by GAPDH.

We have replaced the immunoblots that were overexposed or dirty with new sets. We have also repeated some of the experiments to rectify the inconsistencies with sample loading (Figures 3E, 4B, Supplementary Figure 3B).

In Fig 4A can you show that its K48-linked polyubiquitination using K48-linked specific antibody?? Can you also show that there isn't any K63-linked polyubiquitination using the K63-linked specific antibody?

As suggested by the reviewer, we have now repeated the experiment shown in Figure 4A and performed Western blot analysis using K48 and K63-linked ubiquitin specific antibodies. The results show that the BRE overexpression affects K-48 linked poly-ubiquitylation of CDC25A and not K-63 linked polyubiquitylation. These immunoblots are shown in Supplementary Figure 6 and the results are described on page 11, end of 2nd paragraph. The experimental details are

described in the Methods section.

Explain this sentence further “Involvement of BRCC36 in selective cleavage of K63-linked polyubiquitin further explains its inability to participate in BRE-mediated CDC25A deregulation in response to DNA damage.”

We have now explained why BRCC36 may not contribute to BRE-mediated CDC25A deregulation and changed the text to “K48 polyubiquitin is the major signal for protein degradation, while K63 ubiquitin fulfills other roles such as endocytic trafficking, protein localization after DNA damage response, and activation of the nuclear factor- κ B and T-cell receptor pathways in mammalian cells. This suggests that BRCC36 may not play a role in BRE-mediated CDC25A stability” (Page 12, 2nd paragraph). We have cited the relevant references.

Does the addition of USP7 catalyze deubiquitination of ubiquitinated CDC25A? This should be assessed and added to Fig 4A. What about catalytically inactive USP7? Figs 4B and 4C why do the levels of USP7 fluctuate (lanes 1-4)?

We have examined the effect of USP7 (WT and catalytically inactive) overexpression on polyubiquitylation of CDC25A. Our results show that overexpression of WT USP7 significantly reduces polyubiquitylated CDC25A. In contrast, polyubiquitylated CDC25A is detected on the immunoblot when catalytically inactive USP7 with a mutation (C223S) in the active site is overexpressed. These new results are now shown in Figure 4C.

We have sometimes observed fluctuation in USP7 levels but do not understand the reason of the variability. We have replaced the immunoblot shown in Figure 4B with a new blot with similar USP7 expression.

Describe sequence details of the BRE/BRC45 USP7 binding motifs.

We have included the sequence details on page 15, beginning of 2nd paragraph.

Which USP7 domain is mediating interaction? This should be demonstrated using the individual domains. Typically substrate proteins interact with the USP7-TRAF domain. In this case if BRE/BRC45 contains the binding motif, how does USP7 catalyze deubiquitination of CDC25A? Describe the mechanistic details.

We have constructed two different deletion mutants of USP7 and performed co-immunoprecipitation to identify the region of USP7 that interacts with BRE. Our results shown in Supplementary Fig 9G, suggest that BRE interacts with USP7 via the N-terminal TRAF domain of USP7 (page 15, end of 2nd paragraph). Our results shown in Supplementary Figure 9D suggest that USP7 and CDC25A interaction is dependent on BRE. This makes it difficult to demonstrate whether USP7 directly binds to CDC25A and which domain of USP7 interacts with CDC25A. It is possible that both BRE and CDC25A bind to the TRAF domain of USP7 but the interaction between CDC25A and USP7 is facilitated by the binding of BRE to USP7. We propose that the formation of the CDC25A:BRE:USP7 complex brings USP7 close to CDC25A allowing its subsequent deubiquitylation. We have discussed this possibility in the manuscript on Page 20, end of 2nd paragraph).

There is no discussion as to why there are 3 USP7 interaction sites within BRE/BRC45. Are all

required for stabilization of CDC25A?

As suggested by the reviewer, we have now discussed the significance of 3 USP7 binding motifs within BRE. Because BRE constitutively binds to CDC25A and any one of the three USP7 binding motifs is sufficient to interact with USP7, we predict that any one of the USP7 will be sufficient for stability of CDC25A. This is included in the text (page 20, 2nd paragraph, line 6).

Does CDC25A interact directly with USP7 in the absence of BRE/BRCC45?

We have now examined the effect of BRE knockdown on the interaction between CDC25A and USP7. As shown in Supplementary Fig 9D, the interaction between CDC25A and USP7 is disrupted in the absence of BRE. These results are described in the text on page 15, 1st paragraph, line 6.

What other known BRE/BRCC45 interacting proteins are components of this novel complex?

Does BRCA1 still interact with BRE/BRCC45 when its in the BRE:USP7:CDC25A complex?

To identify other components of the BRE:USP7:CDC25A complex, we have immunoprecipitated CDC25A and examined the pulldown for the presence of known BRE interacting proteins such as BRCC36, RAP80, MERIT40 and BRCA1. Our results suggest that MERIT40 is the only other protein present in the BRE:USP7:CDC25A complex. These results are described in the text on Page 15 (end of 1st paragraph) and shown in Supplementary Fig 9E and 9F.

The absence of USP7 in the co-IP in Supplementary Fig 7C needs further comment from the authors. It looks like the BRE/BRCC45 IPed levels are similar in lanes 2 and 4 so why is USP7 missing from lane 2.

We agree with the reviewer's comment that in the original Supplementary Figure 7C (new Supplementary Figure 9C) we did not detect USP7 although comparable levels of BRE were co-immunoprecipitated in lanes 2 and 4. We attribute this to technical difficulties in detecting co-immunoprecipitated USP7 when both USP7 and BRE were expressed at endogenous levels (lane 2). We have explained this in the text on page 15 (1st paragraph). When we performed similar co immunoprecipitation using BRE antibodies, we could detect USP7 on the immunoblot (Figure 4D, lane 2) suggesting that these proteins do interact even when expressed at endogenous levels.

Are the same effects on the levels of CDC25A seen with a USP7 inhibitor? What about a USP7 active site mutant?

We have examined the effects of USP7 chemical inhibitor (P5091) and found it to induce CDC25A degradation in cells with BRE overexpression. These results are shown in Supplementary Figure 8 and described in the text on Page 13, 3rd paragraph.

We did not perform the experiment with USP7 active site mutant as suggested by the reviewer because when we examined the effect of WT USP7 overexpression (to be used as control), CDC25A was undetectable as shown below:

This effect of USP7 on CDC25A levels contrasts with the effect of BRE overexpression. Interestingly, this effect is independent of DNA damage. Because USP7 has numerous potential substrates and the fact that we did not observe its direct interaction with CDC25A, the effect of USP7 overexpression is likely to be indirect. Therefore, we did not include this result in the manuscript as it is not relevant to the role of BRE on CDC25A stability. It will be interesting in future to investigate the details about the regulation of endogenous CDC25A by USP7 overexpression. Based on our findings shown in Figure 4C, CDC25A is polyubiquitylated in the presence of active site mutant USP7 and is therefore expected to be degraded. In these experiments, polyubiquitylated CDC25A is not observed when USP7 is overexpressed, which apparently contradicts our observation described above. We attribute this difference to the fact that the experiments described in Figure 4C were performed in the presence of MG132, which may stabilize some other regulator of CDC25A. Alternatively, USP7 overexpression may also affect CDC25A transcription.

What happens to levels of CDC25A with overexpressed USP7?

As explained above, when USP7 is overexpressed, the endogenous CDC25A goes down to undetectable levels.

Is BRE (FLΔ123) still active in the BRISC or Rap80 complexes?

Given the essential role of BRE in maintaining the stability of the BRISC complex components such as RAP80 and BRCC36, we tested the activity of USP7-binding deficient BREΔ123 by examining its ability to maintain the stability of these protein. We knocked down endogenous BRE and reconstituted with WT or Δ123 BRE. As shown in Supplementary Figure 10 (and described in the text on Page 16, 2nd paragraph), while knockdown of BRE destabilized RAP80 and BRCC36, levels of both proteins were restored in the presence of WT and Δ123 BRE. This suggests that the Δ123 BRE mutant is fully active and complements the function of full length BRE.

It would be nice to see a positive control in the siUSP7 studies. Do the levels of known USP7 substrates also decrease?

We have examined the levels of HDM2 protein (a direct target of USP7) as a control and included the immunoblots in Figures 4B, and 4C.

Minor points

There are several instances of very short sentences (for example page 4 line 70) which should be expanded or incorporated into the previous sentences.

We have rectified this error by incorporating the short sentences into the previous sentences.

Need a better, more comprehensive introduction on the functions of BRE/BRCC45 that what is currently found on page 4. In addition, these functions should be incorporated into the discussion.

We have now included additional information about BRE functions on Page 4 as well as in the Discussion section on page 19, 2nd paragraph.

Reviewer #2

Major points

1. A more thorough time course with one cell line for BRE knockdown should be shown (e.g. every 30 min after IR for 4 hr for the lines in supplemental, or every 1hr for the line in the main figure). Given the marked changes between early time points after IR and a later time point, I found the effects of BRE depletion to not be entirely convincing.

As suggested by the reviewer, we have now performed a detailed time course experiment (every 1 hour) for MCF7 cells and replaced previous experiment with fewer time points. We observed a more rapid depletion of CDC25A (significant reduction in 1 hr compared to 3 hr) after BRE knockdown using two different siRNAs. The results are shown in Figure 3D.

2. The degree of rescue that is supported by BRE or CDC25A overexpression per se is not well described, apart from the fact that clones could be isolated. I found this to be a weakness of the study. I recognize that this is somewhat difficult, given the incomplete penetrance of CRE activation. However, could an inducible shRNA to BRCA2 (or BRCA1) be used to monitor cell proliferation over time in BRE and CDC25A overexpressed cells, either with the mouse ES or MCF7 models, as in this study (PMID: 23337117)? Namely, is there a resistance to BRCA depletion at early divisions with BRE and/or CDC25A overexpression, or do most of the cells die in the first few days, but then a resistant set of clones grow out. This is an important distinction, which relates to the paradox described in the Discussion regarding the ATR-inhibition work.

The reviewer has raised a very valid and important point regarding the role of BRE in supporting cell viability and growth. To address the concern regarding the degree of rescue of cell viability by BRE, we have now examined the effect of its overexpression on the proliferation of MCF7 cells after BRCA2 knockdown. MCF7 cells grow poorly in response to BRCA2 knockdown. As shown in Figure 1F, when we examined the effect of BRE overexpression on growth of MCF7 cells after BRCA2 knockdown, cell proliferation was found to be comparable to the wild-type cells at initial time points (day 1-5). At later time points (after day 5), the cell proliferation was significantly reduced compared to the wild-type cells. This suggests that although BRE overexpression contributes to resistance to BRCA2-loss induced growth arrest, not all cells survive or proliferate to the same extent. We have now included this observation in the results

section (page 6, 3rd paragraph) and discussed on page 22, 2nd paragraph. We were unable to perform this experiment with CDC25A because we failed to obtain MCF7 clones that stably overexpressed CDC25A.

Minor point

1. mis-spelled PARP at the end.

We have corrected this error.

REVIEWERS' COMMENTS:

Reviewer #1 (Remarks to the Author):

All of my concerns/queries have been addressed. I find that the manuscript is suitable for publication.

Reviewer #2 (Remarks to the Author):

My concerns have been adequately addressed with new data on a time courses in Fig 1F and 3D.

RESPONSE TO THE REVIEWERS' COMMENTS

Reviewer #1

All of my concerns/queries have been addressed. I find that the manuscript is suitable for publication.

Reviewer #2

My concerns have been adequately addressed with new data on a time courses in Fig 1F and 3D.

We are pleased to see that the reviewers are satisfied with our response to their comments. We would like to sincerely thank the reviewers for their constructive comments and valuable suggestions that have significantly improved the overall quality of the manuscript.